# Time-Varying LoRA: Towards Effective Cross-Domain Fine-Tuning of Diffusion Models

**Zhan Zhuang**[1,2,*]   **Yulong Zhang**[3,*]   **Xuehao Wang**[1]
**Jiangang Lu**[3]   **Ying Wei**[3,†]   **Yu Zhang**[1,†]
[1]Southern University of Science and Technology
[2]City University of Hong Kong   [3]Zhejiang University
12250063@mail.sustech.edu.cn   {zhangylcse, lujg, ying.wei}@zju.edu.cn
{xuehaowangfi, yu.zhang.ust}@gmail.com

## Abstract

Large-scale diffusion models are adept at generating high-fidelity images and facilitating image editing and interpolation. However, they have limitations when tasked with generating images in dynamic, evolving domains. In this paper, we introduce Terra, a novel **T**im**e**-va**r**y**r**ank **a**dapter that offers a fine-tuning framework specifically tailored for domain flow generation. The key innovation of Terra lies in its construction of a continuous parameter manifold through a time variable, with its expressive power analyzed theoretically. This framework not only enables interpolation of image content and style but also offers a generation-based approach to address the domain shift problems in unsupervised domain adaptation and domain generalization. Specifically, Terra transforms images from the source domain to the target domain and generates interpolated domains with various styles to bridge the gap between domains and enhance the model generalization, respectively. We conduct extensive experiments on various benchmark datasets, empirically demonstrate the effectiveness of Terra. Our source code is publicly available on https://github.com/zwebzone/terra.

## 1 Introduction

Recently, text-to-image diffusion models [38, 47, 48, 45] have revolutionized computer vision by synthesizing high-quality, creative images. Those models provide a user-friendly method for generating images through text prompts. Furthermore, with advancements in fine-tuning techniques of diffusion models [4], users can easily customize [83], edit [27], and interpolate [88, 80, 7] images. A common approach involves using a low-rank adapter (LoRA) [25] to fine-tune diffusion models with a few images to generate customized images. This inspires a generation-based approach to address a fundamental and classical problem in machine learning known as domain shift.

Domain shift is commonly studied in the cross-domain learning [70, 82, 61] with two settings: unsupervised domain adaptation (UDA) [40, 12, 94], which aims to transfer knowledge from a source domain to a target domain, and domain generalization (DG) [89, 64], which focuses on training a model on source domains and then generalizing to unseen target domains. Prior methods [91, 15, 68, 90, 73] have demonstrated the effectiveness of image translation and interpolation on the learning paradigms based on mixup [79, 60], generative adversarial networks [16, 92], and diffusion models [24, 36]. Considering the impressive capabilities of diffusion models and the efficiency of fine-tuning techniques like LoRA, it is natural to extend them to generate domain flow, which generates intermediate domains and bridges the source and target domains, as illustrated in Fig. 1(b).

---

[*]Equal contribution.
[†]Corresponding authors.

38th Conference on Neural Information Processing Systems (NeurIPS 2024).

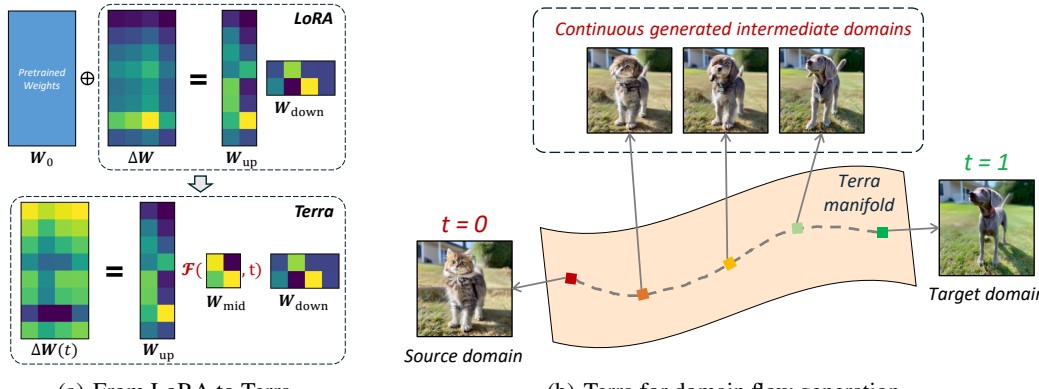

(a) From LoRA to Terra

(b) Terra for domain flow generation

Figure 1: Illustration of the proposed Terra.

However, previous methods [36, 80] require multiple LoRAs to customize multiple domains, since a single LoRA cannot effectively express knowledge of multiple domains with a plugin [77]. To address this limitation, as illustrated in Fig. 1(a), we propose a **T**ime-va**r**ying low-**r**ank **a**dapter (Terra), which offers a framework for gradual domain transferring by constructing a continuous parameter manifold. Instead of training multiple LoRAs for different domains, Terra maintains the parameter efficiency. To this end, inspired by the perspective of dynamic flows [75], Terra introduces a time variable $t$ for each domain and incorporates a square matrix that varies with time $t$ within the original low-rank structure.

As depicted in Fig. 1(b), Terra enables the use of different time values $t$ for various intermediate domains. Consequently, Terra can generate intermediate images that are natural and smooth when morphing in image pairs, subjects, and styles. For UDA tasks, we generate target samples and transform the source samples into the target domain to form an expanded source domain. Due to the smaller domain shifts, transferring from the expanded source domain to the target domain can improve the performance of existing UDA methods. For DG tasks, we interpolate among all source domains to generate images in various styles. Then, the generated samples are combined with the source domain images to improve the performance of existing DG methods.

In summary, our contributions are four-fold:

- We introduce Terra, a novel framework that integrates a square matrix with a time variable $t$ into the original low-rank structure, facilitating effective and flexible knowledge sharing across different domains while maintaining parameter efficiency.
- We provide a theoretical analysis of the expressive power of Terra, comparing it to LoRA.
- We demonstrate the application of Terra in image transformation and generation for UDA tasks and image interpolation for DG tasks via Terra, respectively.
- Extensive experiments validate the effectiveness of Terra across various tasks, including generative interpolation, unsupervised domain adaptation, and domain generalization.

## 2 Related Work

**Fine-Tuning of Text-to-Image Diffusion Models.** The impressive performance of diffusion models [24, 55] has sparked a surge of interest in text-to-image generation tasks. As the demand for personalized content synthesis grows [83], pioneer works such as Textual Inversion [11] and Dream-Booth [49] have proposed optimized text embedding and full fine-tuning frameworks to generate subject images with limited reference samples. Recently, several parameter-efficient methods for fine-tuning diffusion modules have been proposed, including adapters [54], LoRA [17, 50, 52], singular value decomposition on weight matrices [20], subsets of cross-attention [56, 32], and image prompt adapter [81, 76, 37, 66]. Among those methods, several have been developed to address the challenges of multi-concept generation [32, 20, 17] and natural image interpolation [62, 30, 80, 88]. Different from those methods, Terra focuses on generation and interpolation within domain flows.

**Domain Adaptation and Generalization.** UDA [74, 34, 13, 67, 87, 63] is designed to address the challenge of adapting models trained on labeled source domains to unlabeled target domains. The central premise of UDA methods is to learn domain-invariant features that minimize the domain gap. UDA approaches primarily fall into two branches: discrepancy-based methods [34, 72, 93, 19] and adversarial-based methods [13, 46, 85]. Conversely, DG [64, 89] seeks to train models that could generalize well to unseen target domains using multiple source domains. Effective DG methods, such as SWAD [5] and SAGM [65] enhance the generalization by identifying and leveraging flatter minima of training losses landscapes. However, the performance of UDA and DG methods can be constrained by the availability of training data. To address this limitation, recent data augmentation techniques [73, 71, 84, 36] have been developed to improve the transfer effects of UDA and DG methods. Those methods can be categorized into feature-level [71, 95, 42] and image-level methods [73, 84, 36, 22], which enhance transfer performance through the transformation or generation of auxiliary samples at the feature and image levels. For instance, MSGD [71] and GGF [95] use intermediate domains to gradually reduce the domain shift between the source and target domains, while BDG [73] employs pairs of cross-domain generators to synthesize domain-specific data based on the other domains. Additionally, CDGA [22] leverages the latent diffusion model to generate synthetic samples across domains and Domaindiff [36] trains LoRAs for each source domain to conduct domain fusion.

## 3 Methodology

### 3.1 Preliminary

LoRA [25] uses two low-rank matrices, $\boldsymbol{W}_{\text{down}} \in \mathbb{R}^{r \times n}$ and $\boldsymbol{W}_{\text{up}} \in \mathbb{R}^{m \times r}$, where $r \ll \min(m, n)$, to compute the weight matrix updates $\Delta \boldsymbol{W} = \boldsymbol{W}_{\text{up}} \boldsymbol{W}_{\text{down}} \in \mathbb{R}^{m \times n}$. The forward pass of the new weights changes from $h = \boldsymbol{W}_0 \boldsymbol{x}$ to:

$$h = \boldsymbol{W}_0 \boldsymbol{x} + \alpha \Delta \boldsymbol{W} \boldsymbol{x} = \boldsymbol{W}_0 \boldsymbol{x} + \alpha \boldsymbol{W}_{\text{up}} \boldsymbol{W}_{\text{down}} \boldsymbol{x}, \tag{1}$$

where $\alpha$ is a scaling factor for the magnitude of the changes applied to the original weights. Although LoRA is primarily used for fine-tuning large language models, it is also employed in diffusion models for personalizing image generators with limited training samples, targeting specific styles or subjects [49, 52, 80]. The objective function in previous studies is expressed as noise matching:

$$\mathcal{L}(\Delta \boldsymbol{\theta}) = \mathbb{E}_{\boldsymbol{x}_0, \tau \sim \mathcal{U}(1,T), c, \epsilon \sim \mathcal{N}(0,1)} \left[ \left\| \epsilon - \epsilon_{\boldsymbol{\theta}_0 + \Delta \boldsymbol{\theta}} \left( \boldsymbol{x}_\tau, \tau, e(c) \right) \right\|_2^2 \right], \tag{2}$$

where $\boldsymbol{\theta}_0$ and $\Delta \boldsymbol{\theta}$ denote the parameters of the text-to-image diffusion model and LoRA, respectively. The function $e$ denotes the text encoder, and $c$ corresponds to the text prompt. During the forward diffusion process, the variable $\boldsymbol{x}_\tau$ is obtained by gradually adding noise to the initial image $\boldsymbol{x}_0$ using the equation $\boldsymbol{x}_\tau = \sqrt{\bar{\alpha}_\tau} \boldsymbol{x}_0 + \sqrt{1 - \bar{\alpha}_\tau} \epsilon$. Here $\alpha_\tau$ follows a decreasing schedule, and $\bar{\alpha}_\tau$ is calculated as the cumulative product of $\alpha$ values up to timestep $\tau$. In the objection function, the timestep $\tau$ is sampled from a uniform distribution $\mathcal{U}(1, T)$, where $T$ denotes the total number of timesteps. And the model is utilized to predict the noise $\epsilon_{\boldsymbol{\theta}_0 + \Delta \boldsymbol{\theta}}$ to estimate the true noise $\epsilon$. After training, the well-trained denoiser $\boldsymbol{\theta}_0 + \Delta \boldsymbol{\theta}$ can denoise noises and generate images within a few sampling steps.

### 3.2 Terra: Time-Varying Low-Rank Adapter

To address the need for fine-tuning diffusion models across multiple domains while maintaining the parameter efficiency, we propose the Terra, as depicted in Fig. 1(a). Terra involves constructing a LoRA flow that provides a parameter manifold by incorporating time-varying updates as

$$h(t) = \boldsymbol{W}_0 \boldsymbol{x} + \Delta \boldsymbol{W}(t) \boldsymbol{x} = \boldsymbol{W}_0 \boldsymbol{x} + \boldsymbol{W}_{\text{up}} \mathcal{K}(t) \boldsymbol{W}_{\text{down}} \boldsymbol{x}, \quad \mathcal{K}(t) = \mathcal{F}(\boldsymbol{W}_{\text{mid}}, t) \tag{3}$$

where $\boldsymbol{W}_{\text{mid}} \in \mathbb{R}^{r \times r}$, $t$ is a one-parameter variable, and $\mathcal{F}$ is a time-dependent function. This formulation enables the differentiable evolution of the parameters $\Delta \boldsymbol{W}(t)$ based on a middle time-varying matrix $\mathcal{K}(t)$. A simple form of $\mathcal{F}(\boldsymbol{W}, t)$ is $t\boldsymbol{W} + \boldsymbol{I}$, where $\boldsymbol{I}$ represents an identity matrix. Since $r \ll \min(m, n)$, the parameter difference between Terra and LoRA with the same rank is negligible. Furthermore, by setting the parameter $t$ to 0, Terra will degenerate to LoRA. It is worth noting that the form $\mathcal{F}(\boldsymbol{W}, t)$ here is just one of the possible variations. More forms can be found in Table 5 of Appendix B and a comparison with MoE-based LoRA [69] is provided in Appendix E.

Here, we present a theoretical analysis of the expression power of the proposed Terra. We define $\boldsymbol{I}_r$ as a diagonal matrix with its first $r$ diagonal entries as 1 and the remaining entries as 0. In the following theorem, we prove that Terra can effectively implement two LoRAs for specific downstream tasks by constructing a parameter manifold with reduced parameters.

**Theorem 1.** (The Equivariance between Terra and Multiple LoRAs) *Assume there exist two LoRAs* $\Delta \boldsymbol{W}_A, \Delta \boldsymbol{W}_B \in \mathbb{R}^{m \times n}$ *with ranks of $p$ and $q$, respectively, that effectively solve two specific downstream tasks. Let $k = \max\{\mathrm{rank}([\Delta \boldsymbol{W}_A \ \Delta \boldsymbol{W}_B]), \mathrm{rank}([\Delta \boldsymbol{W}_A^T \ \Delta \boldsymbol{W}_B^T])\}$, where $\mathrm{rank}(\cdot)$ denotes the rank of a matrix. Then, there exists a Terra with $\boldsymbol{W}_{up} \in \mathbb{R}^{m \times k}$, $\boldsymbol{W}_{down} \in \mathbb{R}^{k \times n}$, $\boldsymbol{W}_{mid} \in \mathbb{R}^{k \times k}$, and $\mathcal{K}(t) = t\boldsymbol{W}_{mid} + \boldsymbol{I}_r$, such that the updated matrix $\Delta \boldsymbol{W}(t) = \boldsymbol{W}_{up}\mathcal{K}(t)\boldsymbol{W}_{down}$, can simultaneously solve the two downstream task, that is, we have $\Delta \boldsymbol{W}(0) = \Delta \boldsymbol{W}_A$ and $\Delta \boldsymbol{W}(1) = \Delta \boldsymbol{W}_B$.*

In Theorem 1, the number of trainable parameters of Terra is governed by $|\Theta| = (m+n)k + k^2$, contrasting with that of two LoRAs $|\Theta| = (m+n)(p+q)$. Note that $k$ represents the maximum rank of the matrices obtained by concatenating the row and column spaces of the two LoRA matrices, which is not greater than the sum of the ranks of the two LoRA matrices, *i.e.*, $k \le p + q$.

Drawing inspiration from prior research on the expressive power of LoRA [78], we further demonstrate the expressive power of Terra. Here, we focus on the multi-layer feedforward neural network with identity activation functions, and the analysis can be extended to fully connected neural networks and transformer networks [78]. Assuming that the target models $\bar{f}_A$ and $\bar{f}_B$ for two specific tasks, as well as the frozen model $f_0$, are linear, they can be represented as:

$$\bar{f}_A(\boldsymbol{x}) = \overline{\boldsymbol{W}}_A \boldsymbol{x}, \quad \bar{f}_B(\boldsymbol{x}) = \overline{\boldsymbol{W}}_B \boldsymbol{x}, \quad f_0(\boldsymbol{x}) = \boldsymbol{W}_L \cdots \boldsymbol{W}_1 \boldsymbol{x} = \left(\prod_{l=1}^{L} \boldsymbol{W}_l\right) \boldsymbol{x},$$

where the frozen model has $L$ layers with consistent dimensions. We define the error matrices $\boldsymbol{E}_A := \overline{\boldsymbol{W}}_A - \prod_{l=1}^{L} \boldsymbol{W}_l$, and $\boldsymbol{E}_B := \overline{\boldsymbol{W}}_B - \prod_{l=1}^{L} \boldsymbol{W}_l$, and their ranks as $R_{\boldsymbol{E}_A} = \mathrm{rank}(\boldsymbol{E}_A)$ and $R_{\boldsymbol{E}_B} = \mathrm{rank}(\boldsymbol{E}_B)$. By utilizing Terra $\Delta \boldsymbol{W}(t)$, we can modify the pre-trained frozen model to closely approximate the two target models $\overline{\boldsymbol{W}}_A$ and $\overline{\boldsymbol{W}}_B$. We denote the $d$-th largest singular value of $\boldsymbol{W}$ by $\sigma_d(\boldsymbol{W})$, and the best rank-$r$ approximation [8] of $\boldsymbol{W}$ by $\mathrm{LR}_r(\boldsymbol{W})$. The following theorem presents an upper bound for the approximation error with a rank-$k$ Terra.

**Theorem 2.** (The Expressive Power of Terra) *For each layer $l$, the rank-$k$ Terra has updated matrix $\Delta \boldsymbol{W}(t)_l$, and the function of time-varying matrix is $\mathcal{K}(t)_l = t\boldsymbol{W}_{mid,l} + \tilde{\boldsymbol{I}}$. Assume that all weight matrices of the frozen model $(\boldsymbol{W}_l)_{l=1}^{L}$, $\prod_{l=1}^{L} \boldsymbol{W}_l + \mathrm{LR}_r(\boldsymbol{E}_A)$, and $\prod_{l=1}^{L} \boldsymbol{W}_l + \mathrm{LR}_r(\boldsymbol{E}_B)$ are non-singular for all $r \le k(L-1)$. Then the approximation error satisfies*

$$\min_{\Delta \boldsymbol{W}(t)} \left( \left\| \prod_{l=1}^{L}(\boldsymbol{W}_l + \Delta \boldsymbol{W}(0)_l) - \overline{\boldsymbol{W}}_A \right\|_2 + \left\| \prod_{l=1}^{L}(\boldsymbol{W}_l + \Delta \boldsymbol{W}(1)_l) - \overline{\boldsymbol{W}}_B \right\|_2 \right) \le 2\sigma_{kL+1}^*, \quad (4)$$

*where the $\sigma_{kL+1}^*$ as the $(kL+1)$-th largest singular values obtained by merging the singular values of $\boldsymbol{E}_A$ and $\boldsymbol{E}_B$. Moreover, when $k \ge \left\lceil \frac{R_{\boldsymbol{E}_A} + R_{\boldsymbol{E}_B}}{L} \right\rceil$, the approximation error is zero.*

We compare the approximation errors of Terra and multiple LoRAs with consistent parameter sizes for the above target models. We consider a rank of $2k$ for Terra and two $k$-rank LoRA in both tasks. Prior work [78] establishes an upper bound on LoRA's approximation error as $\sigma_{kL+1}(\boldsymbol{E}_A) + \sigma_{kL+1}(\boldsymbol{E}_B)$. In Theorem 2, we demonstrate that Terra's approximation error bound is $2\sigma_{2kL+1}^*$. Considering the definition of $\sigma^*$, it is evident that our Terra's error bound is not greater than LoRA's.

Terra is capable of cross-domain generative tasks, where samples from different domains possess different $t$'s. In the following sections, we show the use of Terra in three different learning problems.

### 3.3 Warm Up: Constructing Evolving Visual Domains via Terra

In this section, we show the first application of Terra to construct evolving visual domains for generative interpolation between two image domains $\mathcal{D}_S$ and $\mathcal{D}_T$ characterized by the differences in the style or subject, which is the key to apply Terra to UDA and DG.

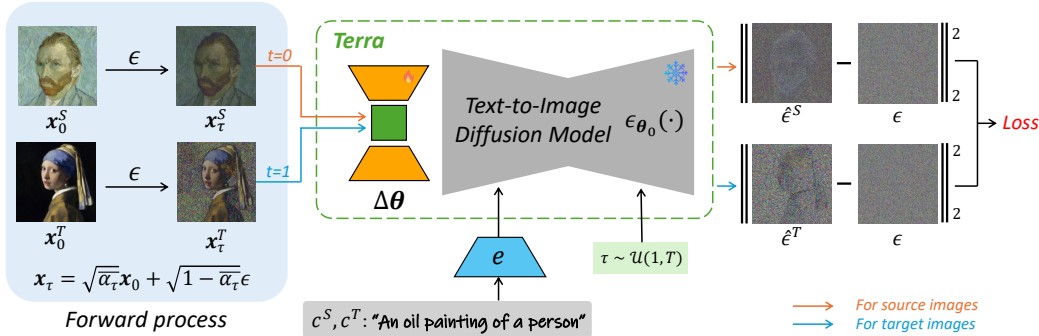

Figure 2: The illustration of the training process of constructing evolving visual domains via Terra.

Our method is different from existing methods [53, 55, 88, 80] that employ direct interpolation between two images on embedding using spherical linear interpolation (*a.k.a* slerp). To accomplish this, Terra incorporates a continuous time variable $t$. Training on the source images involves setting $t$ to 0, yielding the formulation $\Delta \boldsymbol{W}(0) = \boldsymbol{W}_{\text{up}}\mathcal{K}(0)\boldsymbol{W}_{\text{down}}$. Similarly, for the target images, $t$ is set to 1, leading to $\Delta \boldsymbol{W}(1) = \boldsymbol{W}_{\text{up}}\mathcal{K}(1)\boldsymbol{W}_{\text{down}}$. In the context of fine-tuning text-to-image diffusion models, we employ image descriptions to construct prompts for diffusion models, where the corresponding class label is denoted by "A [class]", where "[class]" denotes the placeholder for the class label. Finally, the training objective, as depicted in Fig. 2, is formulated as follows

$$
\begin{aligned}
\mathcal{L}(\Delta \boldsymbol{\theta}) = \mathbb{E}_{\epsilon \sim \mathcal{N}(0,1), \tau \sim \mathcal{U}(1,T)} \big[ & \mathbb{E}_{\boldsymbol{x}_0^S \sim \mathcal{D}_S, t=0} \left\| \epsilon - \epsilon_{\boldsymbol{\theta}_0 + \Delta \boldsymbol{\theta}} \left( \boldsymbol{x}_\tau^S, \tau, e(c^S), t \right) \right\|_2^2 \\
& + \mathbb{E}_{\boldsymbol{x}_0^T \sim \mathcal{D}_T, t=1} \left\| \epsilon - \epsilon_{\boldsymbol{\theta}_0 + \Delta \boldsymbol{\theta}} \left( \boldsymbol{x}_\tau^T, \tau, e(c^T), t \right) \right\|_2^2 \big],
\end{aligned}
\tag{5}
$$

where $\Delta \boldsymbol{\theta}$ represents the parameters of the Terra, $c^S$ and $c^T$ denote the text prompts for the source and target, and $x_0^S$ and $x_0^T$ represent the source and target samples. Formally, we construct evolving visual domains by the following two stages: (1) Fine-tune the parameters of Terra (i.e., $\Delta \boldsymbol{\theta} = W_{up} \cup W_{mid} \cup W_{down}$) using Eq. (5), where the first part with $t = 0$ uses source samples $\mathcal{D}_S$ and the second part with $t = 1$ uses target samples $\mathcal{D}_T$. (2) Generate an intermediate domain by uniformly sampling $t$ from $[0, 1]$ and inputting the text prompt and a random noise into the fine-tuned diffusion model corresponding to domain $t$ for the backward process.

### 3.4 Generation-based Unsupervised Domain Adaptation via Terra

Built on the first application introduced in the previous section, we introduce the second application of Terra in UDA. Under the UDA setting, we have a labeled source domain $\mathcal{D}_S$ and an unlabeled target domain $\mathcal{D}_T$. To alleviate domain shifts, we propose a two-stage framework utilizing a generation-based approach to augment the source domain.

Similar to the construction of evolving domains discussed in Section 3.3, the first stage sets out to train the parameters of Terra that accommodate source domain generation with $t = 0$ and target domain generation with $t = 1$. This enables the generation of target images according to the class labels and transitive source images into the target domain. However, due to the polysemous words on the class labels, directly generating images with the text prompt may cause unexpected results. For example, "mouse" usually refers to a rodent, but in some datasets, it refers to a computer mouse. Therefore, we leverage the source samples to conduct semantic alignment between images and class labels while the unlabeled target domain samples contribute to learning style information for fine-tuning the diffusion model. To achieve this, we adopt the same objective function as Eq. (5), where we set $t = 0$ for source training with the prompt "A [class]" and $t = 1$ for target training with the prompt "An image".

The second stage involves synthesizing a transitive source domain that can benefit the learning of UDA methods, as depicted in Fig. 3(a). We employ two approaches to achieve this. First, we set $t = 1$ to synthesize target samples from Gaussian noises for each category with the corresponding prompt, *i.e.*, "A [class]". Those synthesized samples constitute a generated target domain denoted by $\mathcal{D}_{\hat{T}}$. Second, we transform the source samples into the target domain while preserving semantic

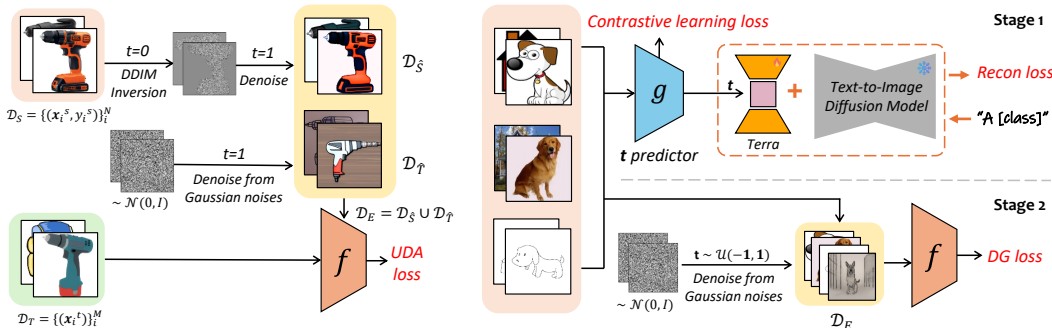

(a) The second stage of our UDA method  (b) The two-stage framework of our DG method

Figure 3: The illustration of the proposed generation-based UDA and DG frameworks via Terra.

information. This is achieved by first setting $t = 0$ and applying DDIM inversion [55] to convert the source images into noise. Then, with setting $t = 1$, we use the diffusion model equipped with Terra to denoise, resulting in the adapted source domain $\mathcal{D}_{\hat{S}}$. After generating images, we combine the adapted source domain and the generated target domain to form a transitive source domain $\mathcal{D}_E = \mathcal{D}_{\hat{S}} \cup \mathcal{D}_{\hat{T}}$. Here the transitive source domain could have a smaller domain gap to the target domain than the original source domain due to the generation process, which could facilitate the knowledge transfer from the transitive source domain to the target domain.

Finally, we conduct transfer learning from the transitive source domain to the target domain by using an existing UDA method. The objective function is formulated as

$$\hat{f}_{uda} = \arg\min_f \frac{1}{|\mathcal{D}_E|} \sum_{(\boldsymbol{x},y)\in\mathcal{D}_E} \ell_{ce}(f(\boldsymbol{x}), y) + \beta \ell_{uda}(\mathcal{D}_E, \mathcal{D}_T), \quad (6)$$

where $\ell_{ce}(\cdot, \cdot)$ denotes the cross-entropy loss, $\beta > 0$ is a trade-off parameter, and $\ell_{uda}(\cdot, \cdot)$ is a transfer loss (*e.g.*, domain discrepancy loss [34, 72, 93] and domain discrimination loss [13, 46, 85]) used to alleviate the domain shift. In this manner, our method can be integrated with any off-the-shelf UDA methods to enhance the transfer performance.

### 3.5 Generation-based Domain Generalization via Terra

In this section, we study the application of Terra to DG problems. Under the DG setting, we have $K$ source domains $\{\mathcal{D}_k = \{(\boldsymbol{x}_i^k, y_i^k)\}_{i=1}^{n_k}\}_{k=1}^K$, where $n_k$ denotes the number of samples in $\mathcal{D}_k$. To enhance the generalization capability, as shown in Fig. 3(b) and detailed as follows, Terra is adopted to synthesize new source domains by interpolating among existing source domains. Consequently, we expect a more generalized learner that well adapts to both existing and synthesized source domains.

In the first stage, to accommodate the various styles exhibited by multiple source domains, we utilize a network $g(\cdot)$ to predict sample-level $\boldsymbol{t}$ for the Terra. The $\boldsymbol{t}$-predictor $g(\cdot)$ aims to generate similar $\boldsymbol{t}$ values for images from the same domain. Moreover, due to the diverse range of styles in the training set, each $\boldsymbol{t} = g(\boldsymbol{x})$ is represented as a vector instead of a scalar value used in previous settings. This allows us to better capture various styles and intra-domain differences. Specifically, we train the network $g(\cdot)$ via contrastive learning and the loss function to be minimized is formulated as

$$\mathcal{L}_{con}(g) = \sum_{k=1}^{K} \sum_{i=1}^{n_k} \left( \sum_{\substack{j=1 \\ j\neq i}}^{n_k} \|g(\boldsymbol{x}_i^k) - g(\boldsymbol{x}_j^k)\|_2 + \sum_{\substack{l=1 \\ l\neq k}}^{K} \sum_{m=1}^{n_l} \max(0, \delta - \|g(\boldsymbol{x}_i^k) - g(\boldsymbol{x}_m^l)\|_2) \right), \quad (7)$$

where $\delta$ is a predefined positive margin and $\|\cdot\|_2$ denotes the Euclidean distance. In Eq. (7), the first term in the sum is to enforce samples from the same domain yield similar outputs, while the second term is to encourage the distance between the outputs corresponding to samples from two domains to be larger than the margin via the hinge loss.

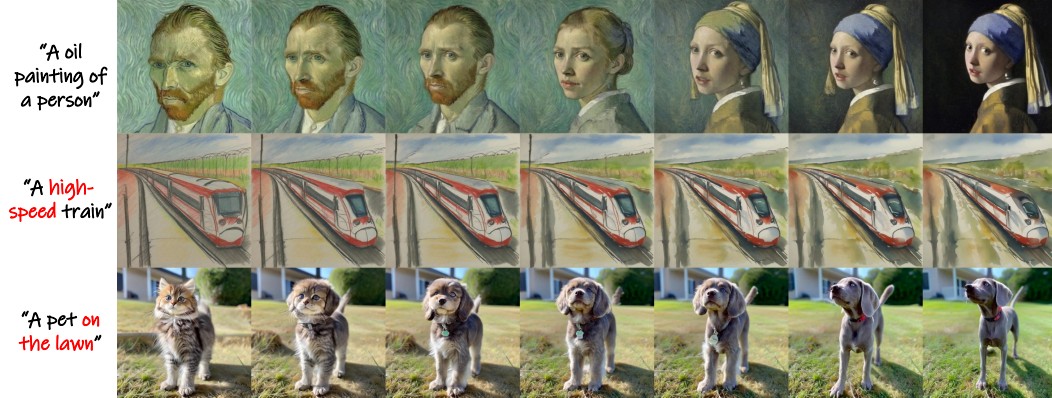

"A oil painting of a person"

"A high-speed train"

"A pet on the lawn"

Figure 4: Qualitative evaluation. The three rows illustrate examples of morphing in image pairs, subjects, and styles, respectively. The text on the left side represents the training prompts, with the red text indicating detailed descriptions used during inference. Additional examples and comparisons with other methods can be found in Appendix C.1.

Upon learning of the $g(\cdot)$, we can obtain $t$'s for all the samples in all the source domains. Then based on $t$'s, we fine-tune the diffusion model using Terra with the prompt "A [class]", and the training objective is formulated as

$$\mathcal{L}(\Delta\boldsymbol{\theta}) = \mathbb{E}_{\epsilon\sim\mathcal{N}(0,1),\tau\sim\mathcal{U}(1,T)}\left[\sum_{k=1}^{K}\mathbb{E}_{\boldsymbol{x}_0\sim\mathcal{D}_k,\boldsymbol{t}=g(\boldsymbol{x}_0)}\|\epsilon - \epsilon_{\boldsymbol{\theta}_0+\Delta\boldsymbol{\theta}}\left(\boldsymbol{x}_\tau,\tau,e(c),\boldsymbol{t}\right)\|_2^2\right]. \quad (8)$$

After fine-tuning, in the second stage, we set $t$ to various values to generate diverse samples for each category with the corresponding prompt. The generated samples could originate from various domains which may be beyond the original source domains $\{\mathcal{D}_k\}_{k=1}^{K}$ but we do not need to identify their specific domains. We combine these generated samples with the original source domain samples to form expanded domains $\mathcal{D}_E$, which can improve the generalization capability of models. The objective function of DG based on Terra is formulated as

$$\hat{f}_{dg} = \arg\min_{f}\frac{1}{|\mathcal{D}_E|}\sum_{(\boldsymbol{x},y)\in\{\mathcal{D}_E\}}\ell_{ce}(f(\boldsymbol{x}),y) + \beta\ell_{dg}(\mathcal{D}_E), \quad (9)$$

where $\beta > 0$ is a trade-off parameter, and $\ell_{dg}(\cdot)$ is a domain generalization loss (*e.g.*, Sharpness-Aware Minimization (SAM)-based loss [10, 5, 65] and representation learning-based loss [1, 13, 2]) used to improve the generalization capabilities. In this manner, our method can be integrated with any off-the-shelf DG methods to enhance their performance.

## 4 Experiments

### 4.1 Experimental Setups

For the UDA experiments, we utilize three benchmark datasets, including *Office31* [51], which consists of 4,110 images from 31 categories across three domains: Amazon (A), Webcam (W), and Dslr (D); *Office-Home* [59], containing 15,588 images from 65 categories across four domains: Art (Ar), Clipart (Cl), Product (Pr), and Real-World (Rw); and *VisDA* [43], featuring 207,785 images from 12 categories across two domains: Synthetic and Real. For the DG experiments, we employ the *PACS* [33], *Office-Home*, and *VLCS* [9] datasets. The *PACS* dataset contains 9,991 images from seven categories across four domains: Art painting (A), Cartoon (C), Photo (P), and Sketch (S), and *VLCS* contains 10,729 images from five categories across four domains: VOC2007 (V), LabelMe (L), Caltech101 (C), and SUN09 (S). The baselines and implementation details are put in Appendix B.

## 4.2 Experiments on Generative Interpolation Tasks

For generative interpolation tasks, we conduct qualitative and quantitative evaluations of our method, focusing on morphing in image pairs, subjects, and styles.

For morphing in image pairs, we train Terra by setting $t = 0$ for the first image and $t = 1$ for the second one with a text prompt "An oil painting of a person". After training, we produce intermediate images by uniformly transitioning $t$ from 0 to 1 with the same text prompt. The experimental results can be found in the first row of Fig. 4. We also provide qualitative comparisons with other baselines in Fig. 8. As can be seen, Terra produces natural and smooth interpolation between two images.

In addition to its ability to perform image morphing, Terra can perform style and subject morphing, a capability that DiffMorpher [80] lacks. Due to page limit, implementation details are put in Appendix B. As shown in the second and third rows of Fig. 4, Terra is capable of generating a sequence of intermediate images as a seamless transition in styles and subjects.

To quantitatively evaluate the quality of the intermediate images and the smoothness of the transition, we utilize the Frechet Inception Distance (FID) [23] and Perceptual Path Length (PPL) [29] metrics, following the setting in DiffMorpher [80]. As shown in Table 1, the quantitative results demonstrate that Terra achieves comparable performance to DiffMorpher and outperforms DGP, DDIM, and LoRA Interpolation. Note that DiffMorpher is specifically designed for morphing by customized techniques such as attention interpolation, adaptive normalization, and a new sampling schedule. Equipped with the customized techniques used in DiffMorpher, Terra is even better than DiffMorhper.

Table 1: Quantitative evaluation of generative interpolation tasks. We evaluate the fidelity and smoothness of the generated intermediate images in terms of FID ($\downarrow$) and PPL ($\downarrow$).

| | image pairs | | styles | | subjects | |
|---|---|---|---|---|---|---|
| | FID | PPL | FID | PPL | FID | PPL |
| DGP (GAN-based) [41] | 223.82 | 1.98 | - | - | - | - |
| DDIM [55] | 176.34 | 1.35 | - | - | - | - |
| LoRA Interp. [80] | 89.37 | 0.91 | 256.64 | 1.02 | 194.17 | 1.24 |
| DiffMorpher [80] | 78.26 | 0.77 | - | - | - | - |
| Terra (ours) | 62.25 | 0.95 | **187.88** | **0.32** | **181.85** | **0.72** |
| Terra+DiffM. | **44.80** | **0.72** | - | - | - | - |

Table 2: Transfer accuracies (%) on the *Office-Home* and *VisDA* datasets under UDA setting. The best performance is highlighted in bold.

| Method | Office-Home | | | | | | | | | | | | | VisDA |
|---|---|---|---|---|---|---|---|---|---|---|---|---|---|---|
| | Ar→Cl | Ar→Pr | Ar→Rw | Cl→Ar | Cl→Pr | Cl→Rw | Pr→Ar | Pr→Cl | Pr→Rw | Rw→Ar | Rw→Cl | Rw→Pr | Avg | mean |
| ERM [58] | 44.06 | 67.12 | 74.26 | 53.26 | 61.96 | 64.54 | 51.91 | 38.90 | 72.94 | 64.51 | 43.84 | 75.39 | 59.39 | 51.47 |
| DANN [13] | 52.53 | 62.57 | 73.20 | 56.89 | 67.02 | 68.34 | 58.37 | 54.14 | 78.31 | 70.78 | 60.76 | 80.57 | 65.29 | 79.02 |
| AFN [72] | 52.58 | 72.42 | 76.96 | 64.90 | 71.14 | 72.91 | 64.08 | 51.29 | 77.83 | 72.21 | 57.46 | 82.09 | 67.99 | 74.64 |
| CDAN [35] | 54.21 | 72.18 | 78.29 | 61.97 | 71.43 | 72.39 | 62.96 | 55.68 | 80.68 | 74.71 | 61.22 | 83.68 | 69.12 | 80.74 |
| MDD [86] | 56.37 | 75.53 | 79.17 | 62.95 | 73.21 | 73.55 | 62.56 | 54.86 | 79.49 | 73.84 | 61.45 | 84.06 | 69.75 | 81.10 |
| SDAT [46] | 58.20 | 77.46 | 81.35 | 66.06 | 76.45 | 76.41 | 63.70 | 56.69 | 82.49 | 76.02 | 62.09 | 85.24 | 71.85 | 83.23 |
| MSGD [71] | 58.70 | 76.90 | 78.90 | 70.10 | 76.20 | 76.60 | 69.00 | 57.20 | 82.30 | 74.90 | 62.70 | 84.50 | 72.40 | 84.60 |
| MCC [28] | 56.83 | 79.81 | 82.66 | 67.80 | 77.02 | 77.82 | 66.98 | 55.43 | 81.79 | 73.95 | 61.41 | 85.44 | 72.24 | 83.32 |
| MCC+Terra | 63.49 | 81.51 | 83.46 | **72.52** | 82.89 | **81.25** | **73.20** | 61.66 | 83.16 | 74.36 | 63.45 | 84.41 | 75.45 | 85.39 |
| ELS [85] | 57.79 | 77.65 | 81.62 | 66.59 | 76.74 | 76.43 | 62.69 | 56.69 | 82.12 | 75.63 | 62.85 | 85.35 | 71.84 | 83.40 |
| ELS+Terra | **64.62** | **82.33** | **83.60** | 71.19 | **84.25** | 80.31 | 73.00 | **63.57** | **83.81** | **76.20** | **66.56** | **85.70** | **76.26** | 86.86 |

## 4.3 Experiments on Unsupervised Domain Adaptation

In this section, we evaluate the proposed generation-based UDA method via Terra as introduced in Section 3.4. The comparison results against state-of-the-art UDA methods on the *Office-Home* and *VisDA* datasets are shown in Tables 2. Due to page limit, detailed results for *VisDA* and *Office31* are shown in Tables 7 and 8 of Appendix C.2 and more results with CoVi and PMTrans are shown in Table 10. The standard deviations from three experiments are presented in Appendix D.

As can be seen, our method has achieved significant performance improvements of 4.42%, 3.46%, and 1.07% for ELS on the *Office-Home*, *VisDA*, and *Office31* datasets, respectively, surpassing all the baseline methods. Thus, Terra can serve as a good plugin for existing UDA methods.

The effectiveness of our method can be further verified through the t-SNE [57] visualizations, as depicted in Fig. 5. The adapted source domain $\mathcal{D}_{\hat{S}}$ and the generated target domain $\mathcal{D}_{\hat{T}}$ exhibit a smaller domain discrepancy to the target domain $\mathcal{D}_T$ than the original source domain, thereby reducing the domain gaps. Additionally, Fig. 6 presents example images illustrating the transformation from the source domain to the target domain. It can be observed that the style transfer is achieved while preserving the semantic information and subject shapes.

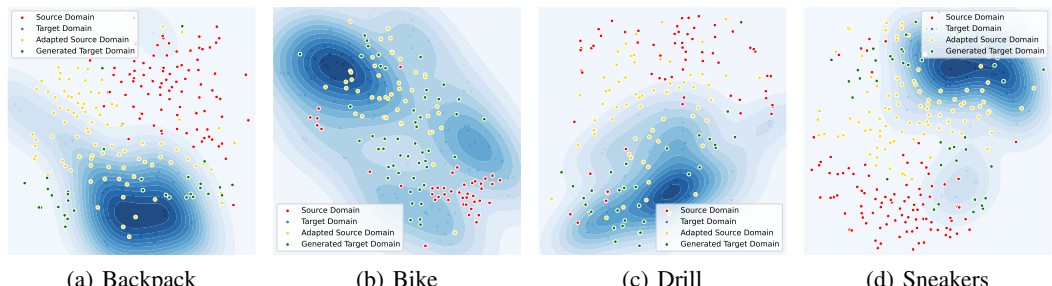

(a) Backpack  (b) Bike  (c) Drill  (d) Sneakers

Figure 5: T-SNE visualization of the source domain, target domain, adapted source domain, and generated target domain in four classes of the Pr→Cl task on *Office-Home* under UDA setting.

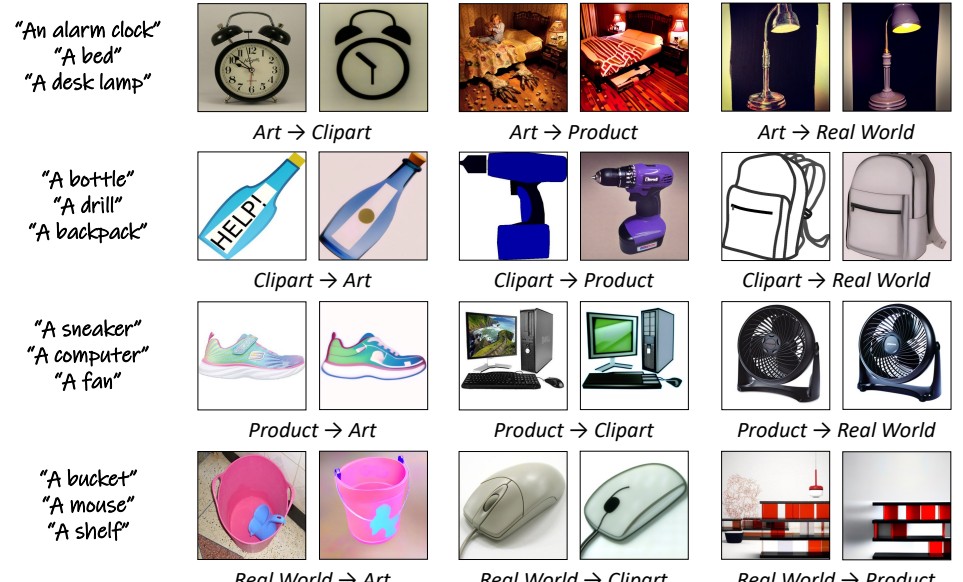

Figure 6: Examples of the source images from $\mathcal{D}_S$ and corresponding adapted images from $\mathcal{D}_{\hat{S}}$ for the *Office-Home* tasks under UDA setting. The text prompts are shown on the left. For instance, the first image pair showcases an image from the Art domain and its corresponding generated image to Clipart domain based on the text prompt "An alarm clock".

Table 3: Ablation studies on the *Office-Home* dataset under UDA setting. The best is in bold.

| Method | Ar→Cl | Ar→Pr | Ar→Rw | Cl→Ar | Cl→Pr | Cl→Rw | Pr→Ar | Pr→Cl | Pr→Rw | Rw→Ar | Rw→Cl | Rw→Pr | **Avg** |
|---|---|---|---|---|---|---|---|---|---|---|---|---|---|
| $\mathcal{D}_S \rightarrow \mathcal{D}_T$ | 57.79 | 77.65 | 81.62 | 66.59 | 76.74 | 76.43 | 62.69 | 56.69 | 82.12 | 75.63 | 62.85 | 85.35 | 71.84 |
| $\mathcal{D}_{\hat{S}} \rightarrow \mathcal{D}_T$ | 61.25 | 78.89 | 80.71 | 68.25 | 79.03 | 75.59 | 66.50 | 60.84 | 80.55 | 73.30 | 65.10 | 84.97 | 72.92 |
| $\mathcal{D}_{\hat{T}} \rightarrow \mathcal{D}_T$ | 58.66 | 80.71 | 80.94 | 69.74 | 80.68 | 78.95 | 69.70 | 54.20 | 81.68 | 71.92 | 56.98 | 82.03 | 72.18 |
| $\mathcal{D}_E \rightarrow \mathcal{D}_T$ | **64.62** | **82.33** | **83.60** | **71.19** | **84.25** | **80.31** | **73.00** | **63.57** | **83.81** | **76.20** | **66.56** | **85.70** | **76.26** |

The ablation studies of ELS+Terra presented in Table 3 show that the best performance is achieved when transferring from the expanded domain $\mathcal{D}_E$ to the target domain $\mathcal{D}_T$, validating the necessity and effectiveness of combining the adapted source domain with the generated target domain. To highlight the design advantages, we conduct a comparison with SDXL's prior knowledge in Appendix C.5.

## 4.4 Experiments on Domain Generalization

In this section, we conduct experiments on the *PACS*, *Office-Home*, and *VLCS* datasets to evaluate the effectiveness of our DG method proposed in Section 3.5. The results presented in Table 4 clearly reveal that our method achieves notable performance improvements across all tasks based on three

Table 4: Accuracies (%) on the *PACS* and *OfficeHome* datasets under DG setting. The best is in bold.

| Method | *PACS* | | | | | *OfficeHome* | | | | |
|---|---|---|---|---|---|---|---|---|---|---|
| | A | C | P | S | **Avg** | Ar | Cl | Pr | Rw | **Avg** |
| MIRO [6] | 87.25 | 76.95 | 97.83 | 77.65 | 84.92 | 67.01 | 55.58 | 78.82 | 81.02 | 70.61 |
| CDGA [22] | 87.30 | 80.90 | 96.60 | 82.50 | 86.80 | 60.50 | 56.50 | 77.10 | 80.60 | 68.70 |
| ERM [58] | 87.00 | 78.23 | 98.05 | 74.35 | 84.41 | 63.41 | 52.61 | 77.20 | 77.63 | 67.71 |
| ERM+DomainDiff [36] | 84.90 | 82.90 | 95.50 | 79.00 | 85.60 | 57.60 | 49.20 | 73.00 | 75.20 | 63.70 |
| ERM+Terra | 89.51 | 79.66 | **98.20** | 78.64 | 86.50 | 65.43 | 53.79 | 78.99 | 80.30 | 69.63 |
| SAGM [65] | 85.72 | 81.13 | 96.59 | 77.46 | 85.23 | 65.55 | 55.09 | 78.68 | 79.39 | 69.68 |
| SAGM+Terra | **91.34** | 82.28 | 96.78 | 80.80 | 87.80 | 66.70 | 56.53 | 79.64 | 81.91 | 71.19 |
| SWAD [5] | 89.67 | 83.13 | 97.48 | 82.78 | 88.27 | 66.08 | 57.37 | 79.58 | 80.49 | 70.88 |
| SWAD+Terra | 91.07 | **83.50** | 98.18 | **84.62** | **89.34** | **68.02** | **58.31** | **80.56** | **82.03** | **72.23** |

(a) P          (b) A          (c) C          (d) S

Figure 7: Visualization of learned time variables on the *PACS* dataset under the DG setting.

state-of-the-art DG methods (*i.e.*, ERM, SWAD, and SAGM). Furthermore, our method outperforms the stable diffusion generation-based method, DomainDiff, which requires training a separate LoRA for each source domain while our method maintains the parameter efficiency with only one single low-rank structure and different $t$ to capture diverse styles.

Additionally, Fig. 7 shows the learned values of $t$. The $t$ predictor assigns distinct $t$ values to each domain, enabling Terra to generate different interpolated images among the source domains based on varying $t$ values. Moreover, the random sampling of $t$ effectively covers the target domain, offering a clearer understanding of the rationale behind our approach that the generated samples may bring useful information for the target domain. We also show some generated images of the expanded domains on the *PACS* dataset in Fig. 10 of Appendix C. As can be seen, using Terra can generate diverse styles of images that are different from the source domains. With the expanded domains, the generalization capability of the source model can be improved.

Besides, we conduct an ablation study on the form of Terra and the dimensionality of $t$ in Appendix B, demonstrating that refining Terra's form can further enhance its expressive power. We also compare Terra with other domain generalization morphing techniques, as shown in Appendix C.4, to verify its effectiveness in expanding source domains for improved generalization.

## 5   Conclusion and Future Works

In this paper, we introduce Terra, a framework that facilitates effective cross-domain modeling through the construction of a continuous parameter manifold. Terra incorporates a time-varying parameter within the manifold of domains, enabling flexible and smooth interpolations. This approach facilitates effective knowledge sharing across different domains by training only a single low-rank adaptor. Additionally, based on the designed generation-based strategies, Terra can serve as a plugin for existing UDA and DG methods to enhance performance. We also theoretically analyze the expressive capabilities of Terra. Extensive experiments demonstrate the superior performance of Terra in a range of tasks. For future works, we aim to extend Terra to cover more settings, including different modalities, larger datasets, and more complex tasks.

## Acknowledgements

This work was supported by NSFC key grant 62136005 and NSFC general grant 62076118.

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

# A Proofs

**Lemma 1.** *Given two matrices* $\mathbf{A}, \mathbf{B} \in \mathbb{R}^{m \times n}$, *and* $\mathrm{rank}(\cdot)$ *denotes the rank of a matrix. Then,* $\mathrm{rank}([\mathbf{A}\ \mathbf{B}]) \leq \mathrm{rank}(\mathbf{A}) + \mathrm{rank}(\mathbf{B})$ *and* $\mathrm{rank}([\mathbf{A}^T\ \mathbf{B}^T]) \leq \mathrm{rank}(\mathbf{A}) + \mathrm{rank}(\mathbf{B})$.

**Lemma 2.** *Given two matrices* $\mathbf{A} \in \mathbb{R}^{m \times n}, \mathbf{B} \in \mathbb{R}^{n \times q}$, *and* $\mathrm{rank}(\cdot)$ *denotes the rank of a matrix. Then,* $\mathrm{rank}([\mathbf{AB}]) \leq \min(\mathrm{rank}(\mathbf{A}), \mathrm{rank}(\mathbf{B}))$.

**Theorem 3.** (Generalized Singular Value Decomposition (GSVD) [39]) *For given matrices* $\mathbf{A}, \mathbf{B} \in \mathbb{R}^{m \times n}$, *let* $\mathbf{C}^T = [\mathbf{A}^T\ \mathbf{B}^T]$ *and denote its rank by* $r = rank(\mathbf{C})$, *there exist orthogonal matrices* $\mathbf{U_A}, \mathbf{U_B} \in \mathbb{R}^{m \times m}$, $\mathbf{Q} \in \mathbb{R}^{n \times n}$ *and* $\mathbf{W} \in \mathbb{R}^{k \times k}$ *so that*

$$\mathbf{U_A}^T \mathbf{A} \mathbf{Q} = \mathbf{\Sigma_A} \left[ \mathbf{W}^T \mathbf{R}, \mathbf{0} \right], \quad \mathbf{U_B}^T \mathbf{B} \mathbf{Q} = \mathbf{\Sigma_B} \left[ \mathbf{W}^T \mathbf{R}, \mathbf{0} \right], \tag{10}$$

$$\mathbf{\Sigma_A} = \begin{bmatrix} \mathbf{I_A} & & \\ & \mathbf{S_A} & \\ & & \mathbf{O_A} \end{bmatrix}, \quad \mathbf{\Sigma_B} = \begin{bmatrix} \mathbf{O_B} & & \\ & \mathbf{S_B} & \\ & & \mathbf{I_B} \end{bmatrix}, \tag{11}$$

*where* $\mathbf{R}$ *is real diagonal contains the nonzero singular values of* $\mathbf{C}$ *in decreasing order,* $\mathbf{\Sigma_A}, \mathbf{\Sigma_B} \in \mathbb{R}^{m \times k}$ *are real non-negative block-diagonal matrices, where* $\mathbf{I_A} \in \mathbb{R}^{r \times r}$ *and* $\mathbf{I_B} \in \mathbb{R}^{k-r-s \times k-r-s}$ *are identity matrices,* $\mathbf{O_A} \in \mathbb{R}^{m-r-s \times k-r-s}$ *and* $\mathbf{O_B} \in \mathbb{R}^{m-k-r \times r}$ *are zero matrices with possibly no rows or no columns, and* $\mathbf{S_A} = [\alpha_{r+1}, \ldots, \alpha_{r+s}]$ *and* $\mathbf{S_B} = [\beta_{r+1}, \ldots, \beta_{r+s}]$. *And we have*

$$1 > \alpha_{r+1} \geq \cdots \geq \alpha_{r+s} > 0, \quad 0 < \beta_{r+1} \leq \cdots \leq \beta_{r+s} < 1, \quad \alpha_i^2 + \beta_i^2 = 1, \quad i \in [r+1, r+s].$$

Indeed, GSVD is a powerful tool in numerical linear algebra and data analysis. It can be seen as an extension of the singular value decomposition (SVD). Notably, when the matrix $\mathbf{B}$ is the identity matrix, the GSVD of matrix $\mathbf{A}$ and $\mathbf{B}$ simplifies to the SVD of matrix $\mathbf{A}$.

**Theorem 1.** (The Equivariance between Terra and Multiple LoRAs) *Assume there exist two LoRAs* $\Delta \boldsymbol{W}_A, \Delta \boldsymbol{W}_B \in \mathbb{R}^{m \times n}$ *with ranks of* $p$ *and* $q$, *respectively, that effectively solve two specific downstream tasks. Let* $k = \max\{\mathrm{rank}([\Delta \boldsymbol{W}_A\ \Delta \boldsymbol{W}_B]), \mathrm{rank}([\Delta \boldsymbol{W}_A^T\ \Delta \boldsymbol{W}_B^T])\}$, *where* $\mathrm{rank}(\cdot)$ *denotes the rank of a matrix. Then, there exists a Terra with* $\boldsymbol{W}_{up} \in \mathbb{R}^{m \times k}$, $\boldsymbol{W}_{down} \in \mathbb{R}^{k \times n}$, $\boldsymbol{W}_{mid} \in \mathbb{R}^{k \times k}$, *and* $\mathcal{K}(t) = t \boldsymbol{W}_{mid} + \boldsymbol{I}_r$, *such that the updated matrix* $\Delta \boldsymbol{W}(t) = \boldsymbol{W}_{up} \mathcal{K}(t) \boldsymbol{W}_{down}$, *can simultaneously solve the two downstream task, that is, we have* $\Delta \boldsymbol{W}(0) = \Delta \boldsymbol{W}_A$ *and* $\Delta \boldsymbol{W}(1) = \Delta \boldsymbol{W}_B$.

*Proof.* Our goal is to find matrices $\boldsymbol{W}_{up}$, $\boldsymbol{W}_{down}$, and $\boldsymbol{W}_{mid}$ to satisfy $\Delta \boldsymbol{W}(0) = \boldsymbol{W}_{up} \boldsymbol{W}_{down} = \Delta \boldsymbol{W}_A$, and $\Delta \boldsymbol{W}(1) = W_{up}(\boldsymbol{W}_{mid} + I) \boldsymbol{W}_{down} = \Delta \boldsymbol{W}_B$.

From Theorem 3, since $k \geq \mathrm{rank}([\Delta \boldsymbol{W}_A^T\ \Delta \boldsymbol{W}_B^T])$, we know GSVD can decompose the two LoRA adapters with a common right generalized singular vectors $\mathbf{X} \in \mathbb{R}^{k \times n}$:

$$\Delta \boldsymbol{W}_A = \mathbf{U_A} \mathbf{\Sigma_A} \mathbf{X}, \quad \Delta \boldsymbol{W}_B = \mathbf{U_B} \mathbf{\Sigma_B} \mathbf{X}. \tag{12}$$

Similarly, we can transpose the matrices of the two LoRA adapters, since $k \geq \mathrm{rank}([\Delta \boldsymbol{W}_A\ \Delta \boldsymbol{W}_B])$, and use GSVD again, then we have a common left generalized singular vectors $\mathbf{Y} \in \mathbb{R}^{m \times k}$:

$$\Delta \boldsymbol{W}_A = \mathbf{Y} \mathbf{Z_A} \mathbf{V_A}, \quad \Delta \boldsymbol{W}_B = \mathbf{Y} \mathbf{Z_B} \mathbf{V_B}. \tag{13}$$

For each matrix $\mathbf{W}$, there exists a pseudo-inverse (*a.k.a.* the Moore-Penrose inverse [44]) $\mathbf{W}^+$ such that $\mathbf{W}\mathbf{W}^+\mathbf{W} = \mathbf{W}$. Then we have a special decomposition of the LoRA adapters:

$$\begin{aligned} \Delta \boldsymbol{W}_A &= \mathbf{U_A} \mathbf{\Sigma_A} \mathbf{X} \\ &= \mathbf{U_A} \mathbf{\Sigma_A} (\mathbf{X} \mathbf{X}^+ \mathbf{X}) \\ &= \Delta \boldsymbol{W}_A \mathbf{X}^+ \mathbf{X} \\ &= \mathbf{Y} \mathbf{Y}^+ \Delta \boldsymbol{W}_A \mathbf{X}^+ \mathbf{X} \triangleq \mathbf{Y} \mathbf{K_A} \mathbf{X}, \end{aligned} \tag{14}$$

where $\mathbf{K_A} \in \mathbb{R}^{k \times k}$. Similarly, we have:

$$\Delta \boldsymbol{W}_B = \mathbf{Y}(\mathbf{Y}^+ \Delta \boldsymbol{W}_B \mathbf{X}^+) \mathbf{X} \triangleq \mathbf{Y} \mathbf{K_B} \mathbf{X}. \tag{15}$$

Assume the SVD of $\mathbf{K_A}$ is of the following form:

$$\mathbf{K_A} = \mathbf{U_K} \mathbf{\Lambda} \mathbf{V_K}, \tag{16}$$

where $\mathbf{U_K}, \mathbf{V_K} \in \mathbb{R}^{k \times k}$ are orthogonal matrices.

We can represent the diagonal matrix $\mathbf{\Lambda}$ as $\mathbf{\Lambda} = \text{diag}(\sigma_1, \sigma_2, \ldots, \sigma_r, 0, \ldots, 0)$. Define $\mathbf{\Lambda}^+$ whose first $r$ rows have $1/\sigma_1, 1/\sigma_2, \ldots, 1/\sigma_r$ on the diagonal, and the product of $\mathbf{\Lambda}$ and $\mathbf{\Lambda}^+$ is a square matrix whose first $r$ diagonal entries are 1 and whose others are 0, i.e. $\mathbf{I}_r$. Then, we can get

$$
\begin{aligned}
\Delta \boldsymbol{W}(t) &= \mathbf{Y}\left(t(\mathbf{K_B} - \mathbf{K_A}) + \mathbf{K_A}\right)\mathbf{X} \\
&= \mathbf{Y}\left(t(\mathbf{K_B} - \mathbf{K_A}) + \mathbf{U_K}\mathbf{\Lambda}\mathbf{V_K}\right)\mathbf{X} \\
&= \mathbf{Y}\mathbf{U_K}\mathbf{U_K}^T\left(t(\mathbf{K_B} - \mathbf{K_A}) + \mathbf{U_K}\mathbf{\Lambda}\mathbf{V_K}\right)\mathbf{V_K}^T\mathbf{V_K}\mathbf{X} \\
&= \mathbf{Y}\mathbf{U_K}\left(t\mathbf{U_K}^T(\mathbf{K_B} - \mathbf{K_A})\mathbf{V_K}^T + \mathbf{\Lambda}\right)\mathbf{V_K}\mathbf{X} \\
&= \mathbf{Y}\mathbf{U_K}\left(t\mathbf{U_K}^T(\mathbf{K_B} - \mathbf{K_A})\mathbf{V_K}^T\mathbf{\Lambda}^+ + \mathbf{I}_r\right)\mathbf{\Lambda}\mathbf{V_K}\mathbf{X} \\
&= \boldsymbol{W}_{\text{up}}\left(t\boldsymbol{W}_{\text{mid}} + \mathbf{I}_r)\right)\boldsymbol{W}_{\text{down}}.
\end{aligned}
\tag{17}
$$

Finally, we can construct the following matrices to prove the theorem:

$$
\boldsymbol{W}_{\text{up}} = \mathbf{Y}\mathbf{U_K}, \quad \boldsymbol{W}_{\text{mid}} = \mathbf{U_K}^T(\mathbf{K_B} - \mathbf{K_A})\mathbf{V_K}^T\mathbf{\Lambda}^+, \quad \boldsymbol{W}_{\text{down}} = \mathbf{\Lambda}\mathbf{V_K}\mathbf{X}.
\tag{18}
$$

$\square$

**Theorem 2.** (The Expressive Power of Terra) *For each layer $l$, the rank-$k$ Terra has updated matrix $\Delta\boldsymbol{W}(t)_l$, and the function of time-varying matrix is $\mathcal{K}(t)_l = t\boldsymbol{W}_{mid,l} + \tilde{\boldsymbol{I}}$. Assume that all weight matrices of the frozen model $(\boldsymbol{W}_l)_{l=1}^L$, $\prod_{l=1}^L \boldsymbol{W}_l + \text{LR}_r(\boldsymbol{E}_A)$, and $\prod_{l=1}^L \boldsymbol{W}_l + \text{LR}_r(\boldsymbol{E}_B)$ are non-singular for all $r \leq k(L-1)$. Then the approximation error satisfies*

$$
\min_{\Delta\boldsymbol{W}(t)}\left(\left\|\prod_{l=1}^L(\boldsymbol{W}_l + \Delta\boldsymbol{W}(0)_l) - \overline{\boldsymbol{W}}_A\right\|_2 + \left\|\prod_{l=1}^L(\boldsymbol{W}_l + \Delta\boldsymbol{W}(1)_l) - \overline{\boldsymbol{W}}_B\right\|_2\right) \leq 2\sigma^*_{kL+1}, \tag{4}
$$

*where the $\sigma^*_{kL+1}$ as the $(kL+1)$-th largest singular values obtained by merging the singular values of $\boldsymbol{E}_A$ and $\boldsymbol{E}_B$. Moreover, when $k \geq \left\lceil \frac{R_{\boldsymbol{E}_A} + R_{\boldsymbol{E}_B}}{L} \right\rceil$, the approximation error is zero.*

*Proof.* We first adopt a similar construction consistently with the prior work [78]:

$$
\boldsymbol{S_A} := \prod_{l=1}^L(\boldsymbol{W}_l + \Delta\boldsymbol{W}(0)_l) - \prod_{l=1}^L\boldsymbol{W}_l \quad \boldsymbol{S_B} := \prod_{l=1}^L(\boldsymbol{W}_l + \Delta\boldsymbol{W}(1)_l) - \prod_{l=1}^L\boldsymbol{W}_l.
\tag{19}
$$

Then, the approximate error can be represented as:

$$
\begin{aligned}
&\min_{\Delta\boldsymbol{W}(t)}\left(\left\|\prod_{l=1}^L(\boldsymbol{W}_l + \Delta\boldsymbol{W}(0)_l) - \overline{\boldsymbol{W}}_A\right\|_2 + \left\|\prod_{l=1}^L(\boldsymbol{W}_l + \Delta\boldsymbol{W}(1)_l) - \overline{\boldsymbol{W}}_B\right\|_2\right) \\
&= \min_{\Delta\boldsymbol{W}(t)}\left(\|\boldsymbol{S_A} - \boldsymbol{E_A}\|_2 + \|\boldsymbol{S_B} - \boldsymbol{E_B}\|_2\right).
\end{aligned}
\tag{20}
$$

Following the prior work, we can also decompose $\boldsymbol{S_A}$ into an accumulation of $\boldsymbol{S_{A_l}}$ as follows:

$$
\begin{aligned}
\boldsymbol{S_A} =&\Delta\boldsymbol{W}(0)_L\prod_{l=1}^{L-1}(\Delta\boldsymbol{W}(0)_l + \boldsymbol{W}_l) + \boldsymbol{W}_L\Delta\boldsymbol{W}(0)_{L-1}\prod_{l=1}^{L-2}(\Delta\boldsymbol{W}(0)_l + \boldsymbol{W}_l) \\
&+ \ldots + \left(\prod_{l=2}^L\boldsymbol{W}_l\right)(\Delta\boldsymbol{W}(0)_1 + \boldsymbol{W}_1) - \prod_{l=1}^L\boldsymbol{W}_l \\
=&\sum_{l=1}^L\underbrace{\left[\left(\prod_{i=l+1}^L\boldsymbol{W}_i\right)\Delta\boldsymbol{W}(0)_l\left(\prod_{i=1}^{l-1}(\boldsymbol{W}_i + \Delta\boldsymbol{W}(0)_i)\right)\right]}_{:=\boldsymbol{S_{A_l}}}.
\end{aligned}
\tag{21}
$$

Similarly, we have $S_B = \sum_{l=1}^{L} S_{B_l}$, $S_{B_l} = \left(\prod_{i=l+1}^{L} W_i\right) \Delta W(1)_l \left(\prod_{i=1}^{l-1} (W_i + \Delta W(1)_i)\right)$.

We select the largest $kL$ largest terms of the singular values of $E_A$ and $E_B$, and we denote there are $p$ values from $E_A$ and $q$ values from $E_B$. To prove the theorem, we need to show the following:

$$\min_{\Delta W(t)} \left(\|S_A - E_A\|_2 + \|S_B - E_B\|_2\right) \leq 2\sigma_{kL+1}^*. \tag{22}$$

Define $E_A' = \mathrm{LR}_p(E_A)$, $E_B' = \mathrm{LR}_q(E_B)$, based on the Eckart-Young Theorem [8], then we have:

$$\left\|E_A' - E_A\right\|_2 + \left\|E_B' - E_B\right\|_2 \leq \sigma_{p+1}(E_A) + \sigma_{q+1}(E_B) \leq 2\sigma_{kL+1}^*. \tag{23}$$

Based on (22) and (23), if we can construct Terra parameter $\Delta W(t)$ to make $S_A = E_A'$ and $S_B = E_B'$, then we will finish the proof. We refer to the SVD of $E_A'$ and $E_B'$ as:

$$E_A' = \mathbf{U_A}\mathbf{\Lambda_A}\mathbf{V_A}, \quad E_B' = \mathbf{U_B}\mathbf{\Lambda_B}\mathbf{V_B}, \tag{24}$$

We introduce $Q_{A,l}$ and $Q_{B,l}$ to divide $E_A'$ and $E_B'$ into $L$ parts:

$$\sum_{l=1}^{L} E_A' Q_{A,l} = E_A' \quad \sum_{l=1}^{L} E_B' Q_{B,l} = E_B', \tag{25}$$

We define $I_{a:b}$ as a diagonal matrix whose diagonal entries from the $a$-th to $b$-th position are 1 and others are 0. Here we define the matrices $(Q_{A,l})_{l=1}^{L}$ and $(Q_{B,l})_{l=1}^{L}$ by:

$$\begin{cases} Q_{A,l} = \mathbf{V_A}\mathbf{I}_{R(l-1)+1:Rl}\mathbf{V_A}^T, Q_{B,l} = \mathbf{0}, & \text{for } Rl < p, \\ Q_{A,l} = \mathbf{V_A}\mathbf{I}_{R(l-1)+1:p}\mathbf{V_A}^T, Q_{B,l} = \mathbf{V_B}\mathbf{I}_{1:Rl-p}\mathbf{V_B}^T, & \text{for } p \leq Rl < p+l, \\ Q_{A,l} = \mathbf{0}, Q_{B,l} = \mathbf{V_B}\mathbf{I}_{R(l-1)-p+1:Rl-p}\mathbf{V_B}^T, & \text{for } p+l \leq Rl. \end{cases} \tag{26}$$

It easy to find that $\mathrm{rank}(Q_{A,l}) + \mathrm{rank}(Q_{B,l}) \leq R$. Based on Lemma 1, we have

$$\mathrm{rank}\left(\begin{bmatrix} E_A'Q_{A,l} & E_B'Q_{B,l} \end{bmatrix}\right) \leq k, \quad \mathrm{rank}\left(\begin{bmatrix} \left(E_A'Q_{A,l}\right)^T & \left(E_B'Q_{B,l}\right)^T \end{bmatrix}\right) \leq k. \tag{27}$$

Now, we show a feasible solution to make $S_A = E_A'$ and $S_B = E_B'$ follows these conditions:

$$\widehat{\Delta W(0)}_l = \left(\prod_{i=l+1}^{L} W_i\right)^{-1} E_A' Q_{A,l}\left(\prod_{i=1}^{l-1}(W_i + \widehat{\Delta W(0)}_i)\right)^{-1}, \quad \text{for all } l \in [L], \tag{28}$$

$$\widehat{\Delta W(1)}_l = \left(\prod_{i=l+1}^{L} W_i\right)^{-1} E_B' Q_{B,l}\left(\prod_{i=1}^{l-1}(W_i + \widehat{\Delta W(1)}_i)\right)^{-1}, \quad \text{for all } l \in [L], \tag{29}$$

$$\mathrm{rank}\left(W_l + \widehat{\Delta W(0)}_l\right) = \mathrm{rank}\left(W_l + \widehat{\Delta W(1)}_l\right) = D, \quad \text{for all } l \in [L-1]. \tag{30}$$

Based on the assumptions of $(W_l)_{l=1}^{L}$, $\prod_{l=1}^{L} W_l + \mathrm{LR}_r(E_A)$, and $\prod_{l=1}^{L} W_l + \mathrm{LR}_r(E_B)$ are non-singular for all $r \leq k(L-1)$ and the Eq. (28) and Eq. (29), it's easy to prove that Eq. (30) is satisfied [78].

Using the Lemma 2 and Eq. (27), we can show $\mathrm{rank}\left(\begin{bmatrix} \widehat{\Delta W(0)}_l & \widehat{\Delta W(1)}_l \end{bmatrix}\right) \leq k$ by

$$\mathrm{rank}\left(\begin{bmatrix} \widehat{\Delta W(0)}_l & \widehat{\Delta W(1)}_l \end{bmatrix}\right)$$

$$= \mathrm{rank}\left(\begin{bmatrix} E_A'Q_{A,l} & E_B'Q_{B,l} \end{bmatrix} \begin{bmatrix} (\prod_{i=1}^{l-1}(W_i + \widehat{\Delta W(0)}_i))^{-1} & \\ & (\prod_{i=1}^{l-1}(W_i + \widehat{\Delta W(1)}_i))^{-1} \end{bmatrix}\right)$$

$$\leq \mathrm{rank}\left(\begin{bmatrix} E_A'Q_{A,l} & E_B'Q_{B,l} \end{bmatrix}\right) \leq k$$

Similarly, we can also get $\mathrm{rank}([\widehat{\Delta W(0)}_l^T \; \widehat{\Delta W(1)}_l^T]) \leq k$. Then, based on Theorem 1, for each layer $l$, we can prove that there exists a Terra can satisfies $\Delta W(0)_l = \widehat{\Delta W(0)}_l$ and $\Delta W(1)_l = \widehat{\Delta W(1)}_l$, thereby completing the proof. $\square$

## B   Baselines and Implementation Details

**Baselines.**   For the generative interpolation tasks, we use DGP [41], DDIM [55], DiffMorpher [80], and LoRA Interpolation for comparison. DGP leverages large-scale pre-trained GAN [3] for image morphing. DDIM means the DDIM inversion and latent interpolation as discussed in [55]. DiffMorpher performs image morphing between two images by interpolating corresponding two LoRAs and latent noises. LoRA Interpolation represents directly training two LoRAs and performing interpolation. For the UDA tasks, we compare with ERM [58] and various UDA methods, including AFN [72], MDD [86], MCC [28], DANN [13], CDAN [35], SDAT [46], ELS [85], and MSGD [71]. We integrate the proposed Terra with the state-of-the-art UDA methods, *i.e.*, MCC, and ELS. For the DG tasks, we compare with ERM [58], MIRO [6], CDGA [22], SWAD [5], SAGM [65], and DomainDiff [36]. Note that previous works [18, 5] have found that ERM is effective in DG and outperforms previous DG methods. Thus, we integrate Terra with ERM and the state-of-the-art DG methods, *i.e.*, SWAD, and SAGM.

**Implementation Details.**   The text-to-image diffusion model used in this paper is the Stable Diffusion XL (SDXL) model [45]. The default resolutions of the generated images are $1024 \times 1024$. The rank of LoRA is set as 16 for generative interpolation tasks and 32 for generation-based UDA and DG tasks. All experiments are conducted on an NVIDIA A100 GPU with three random trials.

For generative interpolation tasks, the training data utilized is sourced from the repository of Diff-morpher[3] and the LoRAs from Hugging Face space "LoRA the Explorer"[4]. The training images can be found in the supplementary materials provided. For morphing in styles, given images in crayon and watercolor styles for training, we set $t = 0$ for training on the crayon images and $t = 1$ for training on the watercolor images, with the prompt being "An image". During the inference phase, by uniformly transitioning $t$ from 0 to 1 and using the text prompt "A high-speed train", the generated results are shown in the second row of Fig. 4. For morphing in subject, given five images of cats and eight images of dogs, we set $t = 0$ for training on the cat images and $t = 1$ for training on the dog images, with the prompt being "A pet". During the inference phase, by uniformly transitioning $t$ from 0 to 1 and using the text prompt "A pet on the lawn", the generated results are shown in the last row of Fig. 4.

For UDA tasks, We generate 50 images per category for the *Office31* and *Office-Home*, and 1000 images per category for the *VisDA* datasets. For images translated from the source domain to the target domain, we scale the long side of each source image to 1024 pixels, adjusting the short side proportionally. Following [46], the *ResNet-50* is used as the backbone on the *Office31* and *Office-Home* datasets, and the *ResNet-101* [21] is used as the backbone on the *VisDA-2017* dataset. The learning rate scheduler follows [13]. For MCC+Terra and ELS+Terra, we follow the settings as the original papers [28, 85].

For DG tasks, we generate 400, 160, and 400 images per category for the *PACS*, *Office-Home*, and *VLCS* datasets, respectively. The dimension of parameter $\boldsymbol{t}$ is set as two, with each dimension sampled from -2 to 2 at intervals of 0.1 to generate diverse samples. We employ *ResNet-50* as the backbone and adopt the same training, evaluation protocols, and hyperparameter search results as outlined in [5, 65, 6]. *ResNet-50* is also used as the backbone for $\boldsymbol{t}$ prediction network $g(\cdot)$.

Table 5: Possible forms of Terra and corresponding differentiable functions.

|  | *General* | | *Diagonal* |
| --- | --- | --- | --- |
|  | Linear | Exponential | Cosine |
| $\mathcal{F}(W, t)$ | $tW + I$ | $\exp(tW)$ | $\cos(tW)$ |
| $\frac{d}{dt}\mathcal{F}(W, t)$ | $W$ | $W \cdot \exp(tW)$ | $-\sin(tW)$ |
| $\mathcal{F}(W, 0)$ | $I$ | $J_r$ | $I$ |

---

[3]https://github.com/Kevin-thu/DiffMorpher/
[4]https://huggingface.co/spaces/multimodalart/LoraTheExplorer

We provide three possible forms of Terra listed in Table 5, *i.e.*, Linear, Exponential, and Cosine. We apply the Linear form of Terra for generative interpolation and UDA tasks and the "Cosine-Sine" form, a variant of "Cosine", for DG tasks. Specifically, the form of "Cosine-Sine" is $\cos(tW)$ on the diagonal of $K(t)$, and $\sin(tW)$ at other positions. To provide more insights, we elaborate on the guiding principles behind the choice of these forms:

○ **Linear**: The $tW + I$ is the simplest form, related to a straight and steady flow, which is sufficient for two domains according to Theorem 1 and 2. Its constant velocity of weight changes ensures smooth morphing and is suitable for simple interpolating between two domains under the UDA setting.

○ **Cosine-Sine**: This form is adopted because of the bounded range and non-linearity of trigonometric functions, preventing image collapse during generation and enabling a complex parameter manifold to capture relationships between multiple domains. We recommend using this form in complex scenarios, such as interpolating multiple domains in DG.

○ **Exponential**: $e^{tW} = I + \sum_{k=1}^{\infty} \frac{t^k}{k!} W^k$, implemented using "torch.matrix_exp", also defines a smooth curve in a high-dimensional manifold. This form is more expressive and suitable for handling multiple domains. Notably, it is related to three types of transformations: scalings, rotations, and shears [14].

Table 6: Evaluation on the dimension of time variable $t$ and Linear form (dim2) of Terra on the *PACS* dataset under the DG setting. The best is in bold.

| Method | *PACS* | | | | |
| --- | --- | --- | --- | --- | --- |
| | A | C | P | S | **Avg** |
| ERM | $87.00_{\pm 0.46}$ | $78.23_{\pm 1.16}$ | $98.05_{\pm 0.06}$ | $74.35_{\pm 3.43}$ | 84.41 |
| ERM+Terra (dim1) | $88.29_{\pm 1.35}$ | $\mathbf{82.36}_{\pm 0.46}$ | $97.53_{\pm 0.28}$ | $73.31_{\pm 1.50}$ | 85.37 |
| ERM+Terra (dim2) | $\mathbf{89.51}_{\pm 0.67}$ | $79.66_{\pm 0.03}$ | $\mathbf{98.20}_{\pm 0.00}$ | $\mathbf{78.64}_{\pm 2.08}$ | **86.50** |
| ERM+Terra (dim3) | $89.26_{\pm 1.70}$ | $81.72_{\pm 2.22}$ | $97.94_{\pm 0.30}$ | $76.11_{\pm 1.11}$ | 86.26 |
| ERM+Terra (Linear) | $87.47_{\pm 0.75}$ | $80.17_{\pm 0.46}$ | $97.85_{\pm 0.28}$ | $77.16_{\pm 1.50}$ | 85.66 |

Empirically, the "Cosine-Sine" form of Terra brings better performance for DG compared with the Linear form according to the results shown in Table 6. As can be seen, ERM+Terra with dimension 2 achieves the best average performance, thus we use 2 as the default dimension for DG tasks.

## C   More Experimental Results

### C.1   Results on Generative Interpolation Tasks

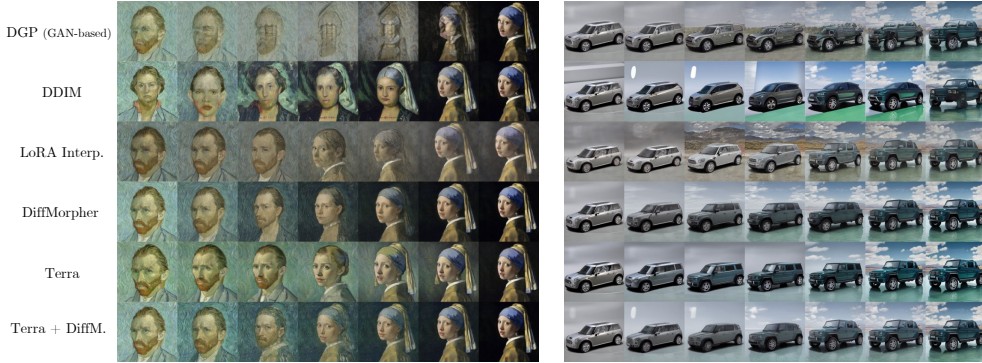

Figure 8: Qualitative results of image morphing using various methods. "Terra + DiffM." integrates Terra with DiffMorpher. As shown, our method generates smooth and natural intermediate images.

The qualitative comparisons of image morphing using various methods are shown in Fig. 8. We perform more qualitative samples of our Terra in Fig. 9. These samples further demonstrate Terra's ability to handle morphing under various scenarios.

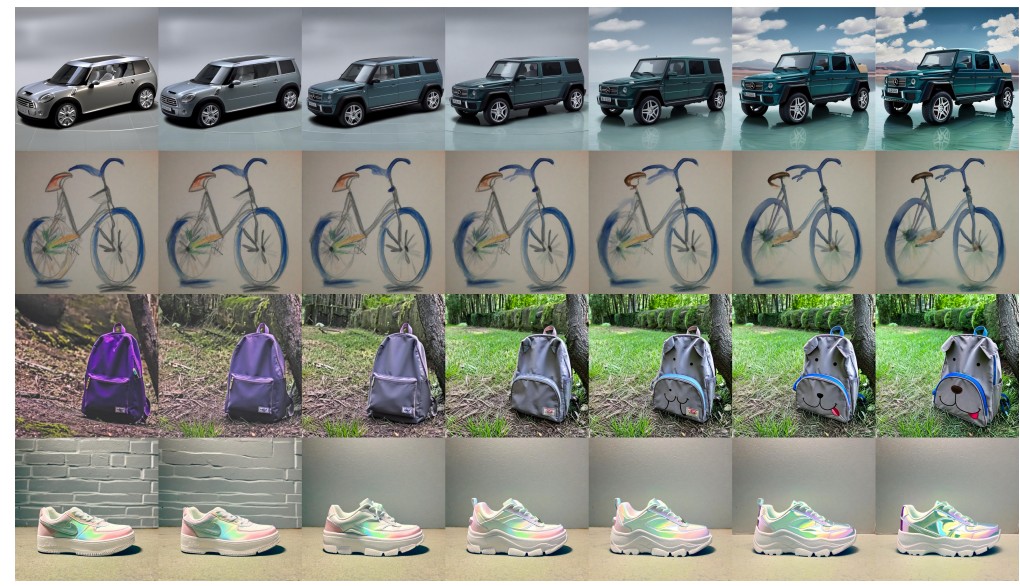

Figure 9: Supplementary samples of qualitative evaluation. The four rows display examples of morphing in image pairs, styles, and objects (purple-to-dog bags, colorful-to-shiny sneakers).

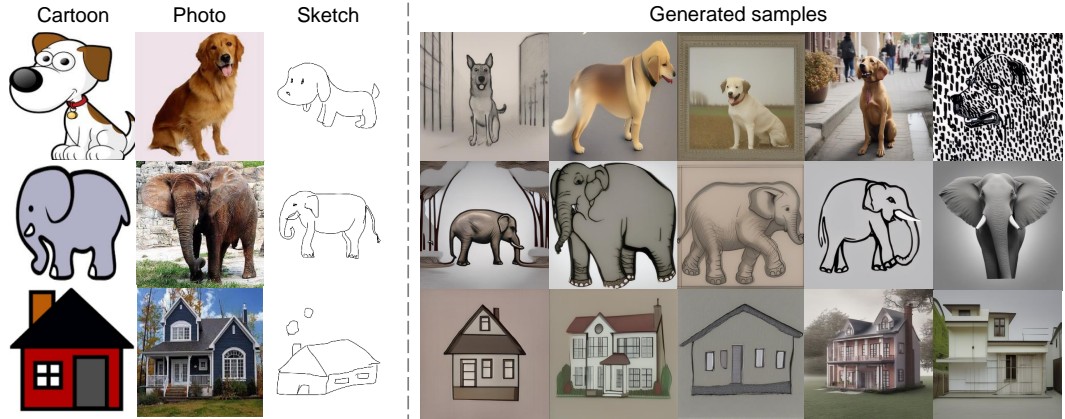

Figure 10: Example images of the expanded domains on the *PACS* dataset under the DG setting.

## C.2  Results on More Datasets

Table 7: Transfer accuracies (%) on the *VisDA* dataset under UDA setting. The best is in bold.

| Method | aero | bicycle | bus | car | horse | knife | motor | person | plant | skate | train | truck | **mean** |
|---|---|---|---|---|---|---|---|---|---|---|---|---|---|
| ERM [58] | 81.71 | 22.46 | 54.08 | **76.21** | 74.83 | 10.69 | 83.81 | 18.71 | 80.88 | 28.66 | 79.66 | 5.98 | 51.47 |
| DANN [13] | 94.75 | 73.47 | 83.46 | 47.91 | 87.00 | 88.30 | 88.47 | 77.18 | 88.16 | 90.05 | 87.21 | 42.26 | 79.02 |
| AFN [72] | 93.13 | 54.76 | 81.03 | 69.74 | 92.36 | 75.88 | 92.11 | 73.83 | 93.16 | 55.55 | **90.48** | 23.63 | 74.64 |
| CDAN [35] | 94.55 | 74.41 | 82.22 | 58.92 | 90.56 | 96.22 | 89.71 | 78.90 | 86.11 | 89.06 | 84.81 | 43.42 | 80.74 |
| MDD [86] | 92.68 | 65.26 | 82.29 | 66.78 | 91.68 | 92.09 | **93.18** | 79.67 | 92.12 | 84.95 | 83.85 | 48.66 | 81.10 |
| SDAT [46] | 94.51 | 83.56 | 74.28 | 65.78 | 93.00 | 95.83 | 89.61 | 80.04 | 90.86 | 91.47 | 84.95 | 54.93 | 83.23 |
| MSGD [71] | **97.50** | 83.40 | **84.40** | 69.40 | **95.90** | 94.10 | 90.90 | 75.50 | **95.50** | **94.60** | 88.10 | 44.90 | 84.60 |
| MCC [28] | 95.26 | 86.14 | 77.12 | 69.98 | 92.83 | 94.84 | 86.52 | 77.78 | 90.26 | 90.98 | 85.68 | 52.52 | 83.32 |
| MCC+Terra | 96.20 | **87.27** | 78.77 | 70.59 | 94.18 | 95.49 | 85.08 | 85.48 | 92.24 | 93.20 | 86.26 | 59.88 | 85.39 |
| ELS [85] | 94.76 | 83.38 | 75.44 | 66.45 | 93.16 | 95.14 | 89.09 | 80.13 | 90.77 | 91.06 | 84.09 | 57.36 | 83.40 |
| ELS+Terra | 95.98 | 87.12 | 81.60 | 70.84 | 95.14 | **96.29** | 88.47 | **87.78** | 94.75 | 94.06 | 86.47 | **63.83** | **86.86** |

Table 8: Transfer accuracies (%) on the *Office31* dataset under the UDA setting. The best is in bold.

| Method | A→W | D→W | W→D | A→D | D→A | W→A | Avg |
|---|---|---|---|---|---|---|---|
| ERM [58] | $77.07_{\pm0.11}$ | $96.60_{\pm0.00}$ | $99.20_{\pm0.00}$ | $81.08_{\pm1.22}$ | $64.11_{\pm0.15}$ | $64.01_{\pm0.11}$ | 80.35 |
| DANN [13] | $89.85_{\pm1.34}$ | $97.95_{\pm0.06}$ | $99.90_{\pm0.08}$ | $83.26_{\pm0.68}$ | $73.28_{\pm0.65}$ | $73.75_{\pm0.39}$ | 86.33 |
| AFN [72] | $91.82_{\pm0.63}$ | $98.77_{\pm0.07}$ | $\mathbf{100.00}_{\pm0.00}$ | $95.12_{\pm0.53}$ | $72.43_{\pm0.50}$ | $70.71_{\pm0.32}$ | 88.14 |
| CDAN [35] | $92.42_{\pm1.75}$ | $98.62_{\pm0.18}$ | $\mathbf{100.00}_{\pm0.00}$ | $91.44_{\pm1.19}$ | $74.61_{\pm0.79}$ | $72.80_{\pm0.45}$ | 88.32 |
| MDD [86] | $93.55_{\pm1.00}$ | $98.66_{\pm0.15}$ | $\mathbf{100.00}_{\pm0.00}$ | $93.92_{\pm0.10}$ | $75.29_{\pm0.68}$ | $73.95_{\pm0.18}$ | 89.23 |
| SDAT [46] | $91.32_{\pm1.83}$ | $98.83_{\pm0.12}$ | $\mathbf{100.00}_{\pm0.00}$ | $95.25_{\pm1.03}$ | $76.97_{\pm0.67}$ | $73.19_{\pm0.34}$ | 89.26 |
| MSGD [71] | $\mathbf{95.50}_{\pm0.50}$ | $99.20_{\pm0.30}$ | $\mathbf{100.00}_{\pm0.00}$ | $95.60_{\pm0.30}$ | $77.30_{\pm0.40}$ | $77.00_{\pm0.50}$ | 90.80 |
| MCC [28] | $94.09_{\pm0.38}$ | $98.32_{\pm0.08}$ | $99.67_{\pm0.09}$ | $94.25_{\pm1.47}$ | $75.89_{\pm0.50}$ | $75.46_{\pm0.20}$ | 89.61 |
| MCC+Terra | $94.55_{\pm0.06}$ | $99.03_{\pm0.06}$ | $\mathbf{100.00}_{\pm0.00}$ | $\mathbf{96.46}_{\pm0.09}$ | $78.64_{\pm0.18}$ | $79.37_{\pm0.12}$ | $\mathbf{91.34}$ |
| ELS [85] | $93.84_{\pm0.51}$ | $98.78_{\pm0.06}$ | $\mathbf{100.00}_{\pm0.00}$ | $95.78_{\pm0.20}$ | $77.72_{\pm0.54}$ | $75.13_{\pm0.16}$ | 90.21 |
| ELS+Terra | $94.09_{\pm0.17}$ | $\mathbf{99.21}_{\pm0.06}$ | $\mathbf{100.00}_{\pm0.00}$ | $96.25_{\pm0.48}$ | $\mathbf{78.67}_{\pm0.28}$ | $\mathbf{79.45}_{\pm0.11}$ | 91.28 |

Table 9: Testing accuracies (%) on the *VLCS* dataset under the DG setting. The best is in bold.

| Method | VLCS | | | | |
|---|---|---|---|---|---|
| | C | L | S | V | Avg |
| MIRO | $98.10_{\pm0.69}$ | $64.05_{\pm1.59}$ | $73.31_{\pm1.78}$ | $76.36_{\pm0.76}$ | 77.95 |
| ERM | $97.76_{\pm1.06}$ | $63.11_{\pm1.50}$ | $72.17_{\pm0.29}$ | $76.56_{\pm2.87}$ | 77.40 |
| ERM+Terra | $98.79_{\pm0.03}$ | $65.54_{\pm1.07}$ | $71.04_{\pm0.45}$ | $77.66_{\pm0.43}$ | 78.25 |
| SAGM | $98.35_{\pm0.36}$ | $65.29_{\pm0.43}$ | $\mathbf{75.22}_{\pm1.09}$ | $79.13_{\pm2.22}$ | 79.50 |
| SAGM+Terra | $\mathbf{99.21}_{\pm0.53}$ | $\mathbf{66.52}_{\pm0.31}$ | $73.95_{\pm0.30}$ | $\mathbf{80.80}_{\pm0.31}$ | $\mathbf{80.12}$ |
| SWAD | $98.74_{\pm0.22}$ | $62.70_{\pm0.43}$ | $74.09_{\pm0.94}$ | $75.64_{\pm1.35}$ | 77.79 |
| SWAD+Terra | $98.94_{\pm0.27}$ | $63.98_{\pm0.02}$ | $73.91_{\pm0.23}$ | $80.19_{\pm0.26}$ | 79.26 |

The complete results on the *VisDA* dataset under the UDA setting are shown in Table 7. The results on the *Office31* dataset under the UDA setting are shown in Table 8, and the results on the *VLCS* dataset under the DG setting are shown in Table 9. As can be seen, Terra is still effective in the two datasets.

## C.3 Results with More Baselines

Moreover, the results with CoVi and PMTrans are shown in Table 10. Notably, Terra consistently improves performance in all tasks with those UDA methods, further verifying the effectiveness of our method.

Table 10: Comparative analysis with two baseline methods on *Office-Home* under UDA setting.

| Method | Ar→Cl | Ar→Pr | Ar→Rw | Cl→Ar | Cl→Pr | Cl→Rw | Pr→Ar | Pr→Cl | Pr→Rw | Rw→Ar | Rw→Cl | Rw→Pr | Avg |
|---|---|---|---|---|---|---|---|---|---|---|---|---|---|
| CoVi | 58.50 | 78.10 | 80.00 | 68.10 | 80.00 | 77.00 | 66.40 | 60.20 | 82.10 | 76.60 | 63.60 | 86.50 | 73.10 |
| CoVi+Terra | 64.56 | 80.65 | 83.36 | 71.45 | 81.03 | 80.77 | 70.83 | 64.86 | 84.07 | 76.76 | 64.19 | 87.18 | 75.81 |
| PMTrans | 82.17 | 91.55 | 92.36 | 89.40 | 92.48 | 92.49 | 87.92 | 80.57 | 92.88 | 88.94 | 82.34 | 94.45 | 88.96 |
| PMTrans+Terra | **83.57** | **93.21** | **92.69** | **89.57** | **92.79** | **93.02** | **89.14** | **82.74** | **93.63** | **89.54** | **83.00** | **94.50** | **89.78** |

## C.4 Comparison of Morphing Works

In addition, for a fair comparison of Terra's effectiveness in expanding source domains that generalize better, we include the comparison against off-the-shelf DG + morphing works on *Office-Home*. That is, we train a LoRA for each domain and adopt LoRA Interp./DiffMorpher to interpolate. The results shown in Table 11 verify the effectiveness of Terra, since Terra interpolates between domains instead of images and thus better models the distributions in two domains.

Table 11: Comparison of morphing works on *Office-Home* using the off-the-shelf method (SWAD) under DG setting. Note that DiffMorpher exhibits lower performance due to the large gap between image pairs, even within the same class.

| Method | Ar | Cl | Pr | Rw | **Avg** |
|---|---|---|---|---|---|
| SWAD | 66.08 | 57.37 | 79.58 | 80.49 | 70.88 |
| +DiffMorpher | 64.06 | 57.43 | 77.91 | 81.04 | 70.11 |
| +LoRA Interp. | 67.23 | 58.06 | 80.09 | 81.33 | 71.68 |
| +Terra | **68.02** | **58.31** | **80.56** | **82.03** | **72.23** |

## C.5 SDXL Prior

To further highlight the design advantages of Terra, we conduct a comparison with data augmentation with SDXL's prior knowledge. Specifically, we design several methods to synthesize data based on the SDXL model and evaluate their effectiveness on UDA tasks:

 (i) SDXL (random): We use the prompt "A [CLASS]" to generate samples for each class, where [CLASS] denotes the placeholder for the label.

 (ii) SDXL (styles): We first use the prompt "Generate 50 prompts describing diverse styles for image generation" to ask GPT-4, and then use the prompt "A [CLASS], an everyday object in office and home, in the style of [STYLE]" to generate samples, where [STYLE] denotes the placeholder for style prompts generated by GPT-4 (*e.g.* "Classic", "Modern").

 (iii) SDXL (target): Based on (ii), we use the name of the target domain (*e.g.* "Clipart") to replace the [STYLE] as the new placeholder for exploring the SDXL prior on the target domain.

 (iv) SDXL (target styles): We use the prompt "Generate 50 prompts describing [TARGET] style for image generation" to ask GPT-4 and obtain more detailed style prompts for synthesis.

 (v) SDXL (selected): Inspired by [26], we use a confidence-based activate learning method to filter out poor-quality and misclassified samples generated in (iv) and select valid samples.

The comparison results on *Office-Home* for UDA are shown in Table 12. Terra outperforms the comparison methods, indicating that despite the boost in accuracy from target style design and active learning, the prior knowledge is insufficient to align with the downstream tasks. This issue can be further mitigated through finetuning with Terra, which demonstrates the design advantages of Terra.

Table 12: Comparison of target-like samples generation by SDXL prior on *Office-Home* based on ELS under UDA setting.

| Method | Ar→Cl | Ar→Pr | Ar→Rw | Cl→Ar | Cl→Pr | Cl→Rw | Pr→Ar | Pr→Cl | Pr→Rw | Rw→Ar | Rw→Cl | Rw→Pr | **Avg** |
|---|---|---|---|---|---|---|---|---|---|---|---|---|---|
| SDXL (random) | 56.88 | 73.64 | 80.38 | 69.18 | 73.64 | 80.40 | 68.93 | 56.54 | 80.38 | 68.93 | 56.54 | 73.64 | 69.92 |
| SDXL (styles) | 55.23 | 77.09 | 80.26 | 68.11 | 77.09 | 80.26 | 68.11 | 55.23 | 80.45 | 68.11 | 55.23 | 77.04 | 70.18 |
| SDXL (target) | 59.70 | 75.51 | 82.26 | 66.67 | 75.51 | **82.26** | 66.67 | 59.70 | 82.26 | 66.67 | 59.70 | 75.51 | 71.04 |
| SDXL (target styles) | 60.76 | 79.52 | 81.68 | 70.95 | 79.52 | 81.68 | 70.95 | 60.76 | 81.68 | 70.95 | 60.76 | 79.52 | 73.23 |
| SDXL (selected) | 61.63 | 79.81 | 82.19 | **71.98** | 79.73 | 81.82 | 71.69 | 61.58 | 82.07 | 72.76 | 62.15 | 80.42 | 73.99 |
| Terra | **64.62** | **82.33** | **83.60** | 71.19 | **84.25** | 80.31 | **73.00** | **63.57** | **83.81** | **76.20** | **66.56** | **85.70** | **76.26** |

# D  Standard Deviations of Experiments

The standard deviations of three random experiments on the *Office-Home*, *VisDA*, and ablation studies under UDA setting are shown in Tables 13, 14, and 15, respectively. Table 16 presents the standard deviations on the *PACS* and *OfficeHome* datasets under DG setting.

# E  Comparison with Other LoRA Variants

In cross-domain learning based on MoLE [69], the process can be viewed as first training LoRAs on different domains separately, followed by training a gating function to integrate the trained LoRAs. Although both MoLE and Terra are designed for diffusion model customization, they differ in several key aspects:

Table 13: The standard deviation of three random experiments on *Office-Home* under UDA setting.

| Method | Ar→Cl | Ar→Pr | Ar→Rw | Cl→Ar | Cl→Pr | Cl→Rw | Pr→Ar | Pr→Cl | Pr→Rw | Rw→Ar | Rw→Cl | Rw→Pr |
|---|---|---|---|---|---|---|---|---|---|---|---|---|
| ERM [58] | 0.25 | 0.26 | 0.42 | 0.17 | 0.20 | 0.15 | 0.07 | 0.17 | 0.05 | 0.34 | 0.33 | 0.01 |
| DANN [13] | 0.44 | 0.72 | 0.38 | 0.02 | 0.30 | 0.39 | 0.58 | 0.47 | 0.59 | 0.84 | 0.14 | 0.51 |
| AFN [72] | 0.16 | 0.30 | 0.06 | 0.23 | 0.31 | 0.14 | 0.32 | 0.15 | 0.02 | 0.19 | 0.18 | 0.22 |
| CDAN [35] | 0.25 | 0.62 | 0.22 | 0.37 | 0.58 | 0.30 | 0.57 | 0.36 | 0.16 | 0.33 | 0.23 | 0.35 |
| MDD [86] | 0.51 | 0.32 | 0.06 | 0.24 | 0.73 | 0.41 | 0.36 | 0.53 | 0.24 | 0.03 | 0.09 | 0.11 |
| SDAT [46] | 0.51 | 0.44 | 0.24 | 0.13 | 0.41 | 0.01 | 1.46 | 0.40 | 0.11 | 0.46 | 0.19 | 0.29 |
| MCC [28] | 0.59 | 0.22 | 0.16 | 0.27 | 0.52 | 0.16 | 0.16 | 0.38 | 0.25 | 0.35 | 0.35 | 0.23 |
| MCC+Terra | 0.21 | 0.11 | 0.14 | 0.25 | 0.28 | 0.18 | 0.18 | 0.29 | 0.25 | 0.15 | 0.06 | 0.11 |
| ELS [85] | 0.83 | 0.45 | 0.38 | 0.08 | 0.46 | 0.19 | 0.39 | 0.39 | 0.08 | 0.02 | 0.44 | 0.05 |
| ELS+Terra | 0.06 | 0.30 | 0.14 | 0.30 | 0.37 | 0.21 | 0.10 | 0.18 | 0.13 | 0.68 | 0.24 | 0.16 |

Table 14: The standard deviation of three random experiments on *VisDA* under UDA setting.

| Method | aero | bicycle | bus | car | horse | knife | motor | person | plant | skate | train | truck |
|---|---|---|---|---|---|---|---|---|---|---|---|---|
| ERM [58] | 9.90 | 2.64 | 3.26 | 2.20 | 1.35 | 3.60 | 1.41 | 1.03 | 1.80 | 3.97 | 0.79 | 0.67 |
| DANN [13] | 0.39 | 1.94 | 0.38 | 2.80 | 0.80 | 3.40 | 0.76 | 0.86 | 0.72 | 2.00 | 0.32 | 2.72 |
| AFN [72] | 0.69 | 3.84 | 1.80 | 2.55 | 1.48 | 2.51 | 0.48 | 2.08 | 2.47 | 3.93 | 1.11 | 1.27 |
| CDAN [35] | 0.38 | 3.72 | 2.55 | 1.36 | 0.53 | 0.52 | 0.14 | 2.58 | 0.67 | 0.49 | 2.61 | 2.43 |
| MDD [86] | 2.40 | 9.46 | 1.18 | 0.66 | 0.85 | 4.01 | 0.65 | 1.81 | 1.21 | 4.30 | 1.58 | 0.33 |
| SDAT [46] | 1.40 | 2.64 | 1.60 | 1.67 | 0.48 | 0.92 | 0.82 | 0.24 | 0.78 | 0.84 | 1.36 | 0.60 |
| MCC [28] | 0.12 | 0.92 | 2.91 | 0.39 | 0.28 | 0.54 | 0.80 | 0.87 | 0.15 | 0.77 | 0.55 | 2.25 |
| MCC+Terra | 0.21 | 0.59 | 0.12 | 0.69 | 0.60 | 0.60 | 0.56 | 0.35 | 0.40 | 0.35 | 0.47 | 0.88 |
| ELS [85] | 0.93 | 1.20 | 1.39 | 0.47 | 0.15 | 0.95 | 1.38 | 0.73 | 1.59 | 1.02 | 1.35 | 0.27 |
| ELS+Terra | 0.34 | 0.79 | 0.41 | 1.31 | 0.06 | 0.39 | 0.85 | 0.43 | 0.92 | 0.64 | 0.26 | 0.19 |

Table 15: The standard deviation of three random experiments of ablation studies of ELS+Terra on *Office-Home* under UDA setting.

| Method | Ar→Cl | Ar→Pr | Ar→Rw | Cl→Ar | Cl→Pr | Cl→Rw | Pr→Ar | Pr→Cl | Pr→Rw | Rw→Ar | Rw→Cl | Rw→Pr |
|---|---|---|---|---|---|---|---|---|---|---|---|---|
| $\mathcal{D}_S \to \mathcal{D}_T$ | 0.83 | 0.45 | 0.38 | 0.08 | 0.46 | 0.19 | 0.39 | 0.39 | 0.08 | 0.02 | 0.44 | 0.05 |
| $\mathcal{D}_{\hat{S}} \to \mathcal{D}_T$ | 0.31 | 0.09 | 0.22 | 0.06 | 0.13 | 0.34 | 0.33 | 0.15 | 0.24 | 0.13 | 0.01 | 0.08 |
| $\mathcal{D}_{\hat{T}} \to \mathcal{D}_T$ | 0.36 | 0.18 | 0.43 | 0.17 | 0.28 | 0.14 | 0.07 | 0.08 | 0.25 | 0.39 | 0.09 | 0.59 |
| $\mathcal{D}_E \to \mathcal{D}_T$ | 0.06 | 0.30 | 0.14 | 0.30 | 0.37 | 0.21 | 0.10 | 0.18 | 0.13 | 0.68 | 0.24 | 0.16 |

Table 16: The standard deviation on the *PACS* and *OfficeHome* datasets under DG setting.

| Method | PACS | | | | OfficeHome | | | |
|---|---|---|---|---|---|---|---|---|
| | A | C | P | S | Ar | Cl | Pr | Rw |
| MIRO [6] | 1.22 | 1.66 | 0.21 | 1.18 | 0.39 | 0.49 | 0.30 | 0.43 |
| CDGA [22] | 1.50 | 1.60 | 0.70 | 0.90 | 1.20 | 0.30 | 0.40 | 0.20 |
| ERM [58] | 0.46 | 1.16 | 0.06 | 3.43 | 0.72 | 0.63 | 0.34 | 0.49 |
| ERM+DomainDiff [36] | 1.60 | 0.00 | 0.00 | 0.90 | 0.40 | 0.60 | 0.60 | 0.90 |
| ERM+Terra | 0.75 | 1.57 | 0.37 | 3.47 | 0.15 | 0.74 | 0.15 | 0.14 |
| SAGM [65] | 0.86 | 1.48 | 0.74 | 2.49 | 0.33 | 0.79 | 0.38 | 0.06 |
| SAGM+Terra | 0.12 | 0.61 | 0.30 | 1.86 | 0.67 | 0.63 | 0.58 | 0.36 |
| SWAD [5] | 0.08 | 0.73 | 0.04 | 0.38 | 0.17 | 0.17 | 0.10 | 0.65 |
| SWAD+Terra | 0.10 | 0.03 | 0.28 | 0.83 | 0.28 | 0.21 | 0.45 | 0.37 |

**Objective**: MoLE focuses on combining multiple pre-trained LoRAs to achieve multi-concept customization, whereas Terra aims to learn a single adapter structure that can capture multiple domains and construct a domain flow for generation.

**Training**: MoLE only optimizes the gating function to preserve the characteristics of trained LoRAs on different domains, whereas Terra participates in the diffusion fine-tuning stage and aims to learn domain-general knowledge and domain-specific knowledge, allowing for control over different domains through a time variable.

**Expressiveness**: MoLE uses a separate gating function for each LoRA layer, which requires entropy-based balancing to resolve conflicts when combining multiple LoRAs. In contrast, Terra achieves domain adaptation through a single time variable $t$, making it more stable. For two-domain interpolation, Terra and MoLE have similar expressiveness. Considering two domains with time variables $t_1$ and $t_2$, we have

$$
\begin{aligned}
\Delta W(\alpha t_1 + (1-\alpha)t_2) &= B\mathcal{K}(\alpha t_1 + (1-\alpha)t_2)A \\
&= (\alpha t_1 + (1-\alpha)t_2)BWA + BA \\
&= \alpha\Delta W(t_1) + (1-\alpha)\Delta W(t_2).
\end{aligned}
\tag{31}
$$

This is equivalent to the linear arithmetic composition in MoLE.

Finally, the relation between MoLE and Terra is similar to that between Gaussian Mixture Model (GMM) and Gaussian Process (GP). GMM composes a complex distribution by multiple Gaussian distributions, and GP is a distribution over functions within a continuous domain (such as time). Analogously, MoLE excels at composition capabilities, while Terra excels at constructing a manifold.

## F  Broader Impact and Ethics Statements

The ability to generate realistic images can be misused to create deepfakes or other deceptive content, potentially leading to misinformation and privacy violations. While our work has the potential to advance the field of PEFT and generation-based cross-domain learning, it is crucial to address the associated risks, particularly in terms of ethical considerations.

## G  Limitation and Failure Cases

Despite showing promising results in data-augmentation-based UDA and DG, Terra has some limitations. Generating images via Terra for data augmentation requires additional storage space. For UDA tasks, we generate target domain samples and transform source domain samples into the target domain, without utilizing Terra's ability to generate intermediate domains. Note that the intermediate domain can be leveraged by using methods in gradual domain adaptation [31], but we have not explored this due to different settings. We leave it for future studies. Additionally, while we have adapted to downstream domains through fine-tuning, our model may still be influenced by the prior of the foundation model to some extent.

We acknowledge that a small number of generated images may exhibit poor quality due to the conflict between SD prior knowledge and the knowledge required for downstream tasks. We showcase some failure cases in Fig. 11. However, the number of those poor-quality images is small, and it does not affect the overall performance of the model.

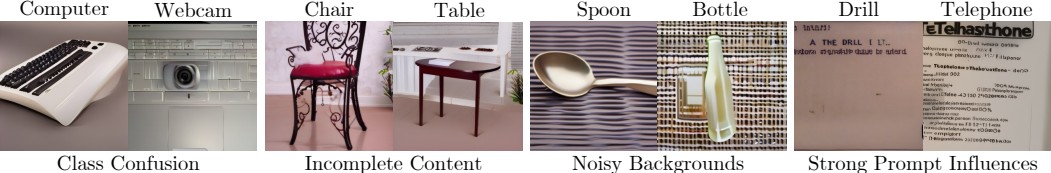

Figure 11: Illustration of failure cases in generated samples.

