# OpenReview forum: "Time-Varying LoRA: Towards Effective Cross-Domain Fine-Tuning of Diffusion Models"
_NeurIPS.cc/2024/Conference — NeurIPS 2024 poster_

### Official Review · Reviewer_STii · 2024-07-02

**Soundness:** 4
**Presentation:** 4
**Contribution:** 3
**Rating:** 8
**Confidence:** 4

**Summary:**

This paper introduces a low-rank adapter, Terra, for effective cross-domain modelling through the construction of a continuous parameter manifold. This approach facilitates knowledge sharing across different domains by training only a single low-rank adaptor. The expressiveness of the model was analyzed theoretically and bounds for approximation error were given. Extensive experiments on various UDA and UG benchmarks demonstrate the effectiveness of the proposed method. Ablation studies verify the robustness of the proposed framework to variations of key components.

**Strengths:**

**(S1)** The proposed Terra is a simple yet effective PEFT method for diffusion models fine-tuning, which successfully generates various images in a customized domain flow.

**(S2)** This paper is well-written and easy to follow. The proposed framework facilitate effective and flexible knowledge sharing across different domains while maintaining parameter efficiency.

**(S3)** Terra involves constructing a continuous parameter manifold using a time variable, with its expressive power theoretically analyzed and smooth interpolation empirically verified.

**(S4)** Terra can serve as a plugin for existing UDA and DG methods to help alleviate domain shifts, achieving state-of-the-art performance on various benchmarks.

**(S5)** Codes are available for reproducibility. (Thanks for providing the codes, resolved some key questions on the implementations of the method)

**Weaknesses:**

**(W1) Qualitative Evaluation**: Although Figure 3 is interesting as it demonstrate how Terra can handle morphing under different scenarios, it is suggested that the authors perform more qualitative samples to evaluate the interpretability of Terra more comprehensively.

**(W2) Comparative analysis**: Although the paper includes extensive comparisons of Terra with two baseline methods in the UDA experiments, it would be beneficial to extend comparisons with recent CNN-based and ViT-based UDA methods, such as CoVi [1] and PMTrans [2].

**(W3) Clarity of Technical Details**: The rationale behind randomly sampling the value of "t" for the DG experiments to generate diverse domains is unclear. While Figure 6 shows that the learned time variables cluster within the existing source domains, the distribution of the target domain is not adequately explained.

[1] Contrastive vicinal space for unsupervised domain adaptation, ECCV 2022.

[2] Patch-mix transformer for unsupervised domain adaptation: A game perspective, CVPR 2023.

**Questions:**

In addition to the above weaknesses, I have the following questions regarding the theoretical part:

**(Q1) Scalability to multiple domains:** To my understanding, Terra can express knowledge of multiple domains (e.g., in the DG setting). However, the paper only demonstrates its equivalence with two LoRAs in Theorem 1. I am curious about the scalability of Terra when applied to more domains or tasks (since this greatly enhances the potential of Terra under real-world scenarios (i.e., multiple environments)).
**(Q2) Comparison with other LoRA variants**: Could the authors provide an analysis of how Terra differs from other LoRA variants, such as MoLE [3], particularly on the expressiveness of these models?

[3] Mixture of LoRA Experts, ICLR 2024.

**Limitations:**

The authors discuss the limitations adequately in the Appendix.

---

> ### Author Rebuttal · Authors · 2024-08-07
>
> #### [W.1] Qualitative Evaluation
>  > We appreciate the reviewer's suggestion. we have included additional qualitative samples in Fig. r2 of the Rebuttal-PDF. These samples further demonstrate Terra's ability to handle morphing under various scenarios.
>
> #### [W.2] Comparative Analysis
> > We thank the reviewer for suggesting additional comparisons. In response, we have conducted experiments with CoVi and PMTrans, and the results are presented in Tab. r4 of the Rebuttal-PDF. Notably, Terra consistently improves performance in all tasks with those UDA methods, further verifying the effectiveness of our method.
>
> #### [W.3] Clarity of Technical Details
>
> > Thank you for your comment. In the DG experiments, we first use contrastive learning loss to train a network $g(\cdot)$ to predict sample-level $t$ for Terra. This approach allows us to better capture inter-domain and intra-domain styles differences. After training, randomly sampling the value of $t$ can generate more diverse samples between domains. Specifically, we use two-dimensional $t$, with each dimension sampled from -2 to 2 at intervals of 0.1 to generate diverse samples. We will include these details in the revision.
> > Moreover, we present the distribution of the **learned time variable** of the target domain in Fig. r4 of the Rebuttal-PDF. As illustrated, the random sampling of $t$ effectively covers the target domain, offering a clearer understanding of the rationale behind our approach that the generated samples may bring useful information for the target domain under the DG setting.
>
> #### [Q.1] Scalability to Multiple Domains
>
> > We appreciate the reviewer's interest in the scalability of Terra. Our theoretical analysis indeed extends to multiple domains or tasks.
> >
> > **Theorem 1 can be generalized to multiple matrices using higher-order generalized singular value decomposition (HO GSVD)** [r1]. Specifically, given K low-rank adapters $\Delta W_k \in \mathbb{R}^{m \times n}$ for K domains/tasks,  we can concatenate the matrices horizontally and vertically to form $H$ and $V$, and let $k = \max\\{rank(H), rank(V)\\}$. Then, similar to Eqs. (14) and (15), based on HO GSVD, each matrix can be exactly factored as $\Delta W_i = Y K_i X$, where $Y \in \mathbb{R}^{m \times k}$, $K_i \in \mathbb{R}^{k \times k}$, and $X \in \mathbb{R}^{k \times n}$. We can then construct $\mathcal{K}(t)$ to satisfy $\mathcal{K}(t_i) = K_i$ for fixed $t_i$. In the case of multiple domains or tasks, we propose using a vector $\symbfit{t}_i$ with a small increase in parameters. The matrix $K_i$ can be constructed using interpolation methods, such as polynomial interpolation or spline interpolation, or non-linear time-varying matrices as those in Table 6.
> >
> > In summary, this approach allows Terra to use one adapter structure to represent multiple LoRAs with fewer parameters. When the domains share knowledge—implying $k$ is small—the required parameters are further reduced. Additionally, for unknown tasks, we can determine the matrices $\mathcal{K}(t_i)$ using least squares or other optimization algorithms [r2], enabling a meta-learning approach.
> >
> >
> >[r1] Ponnapalli SP, Saunders MA, Van Loan CF, Alter O. A Higher-Order Generalized Singular Value Decomposition for Comparison of Global mRNA Expression from Multiple Organisms. PLOS ONE, 2011.
> >
> >[r2] Friedland S, Torokhti A. Generalized rank-constrained matrix approximations. SIAM Journal on Matrix Analysis and Applications, 2007.
>
> #### [Q.2] Comparison with Other LoRA Variants
> > For cross-domain learning based on MoLE, it can be seen as respectively training LoRAs on different domains first and then training a gating function to combine the trained LoRAs. While both MoLE and Terra are designed for customization of diffusion models, they differ in several key aspects:
> > - **Objective**: MoLE focuses on combining multiple pre-trained LoRAs to achieve multi-concept customization, whereas Terra aims to learn a single adapter structure that can capture multiple domains and construct a domain flow for generation.
> > - **Training**: MoLE only optimizes the gating function to preserve the characteristics of trained LoRAs on different domains, whereas **Terra participates in the diffusion fine-tuning stage** and aims to learn domain-general knowledge and domain-specific knowledge, allowing for control over different domains through a time variable.
> > - **Expressiveness**: MoLE uses a separate gating function for each LoRA layer, which requires entropy-based balancing to resolve conflicts when combining multiple LoRAs. In contrast, Terra achieves domain adaptation through a single time variable $t$, making it more stable. For two-domain interpolation, Terra and MoLE have similar expressiveness. Considering two domains with time variables $t_1$ and $t_2$, we have
> > $$
> > \begin{equation}
> > \Delta W(\alpha t_1 + (1-\alpha)t_2) =  B\mathcal{K}(\alpha t_1 + (1-\alpha)t_2)A =(\alpha t_1 + (1-\alpha)t_2)BWA+ BA = \alpha \Delta W(t_1) + (1-\alpha) \Delta W(t_2).
> > \end{equation}
> > $$
> > This is equivalent to the linear arithmetic composition in MoLE. As shown in the response to [Q.1], this conclusion can be extended to interpolation with three or more LoRAs.
> >
> > Finally, the relation between MoLE and Terra is similar to that between **Gaussian Mixture Model (GMM)** and **Gaussian Process (GP)**. GMM composes a complex distribution by multiple Gaussian distributions, and GP is a distribution over functions within a continuous domain (such as time). Analogously, MoLE excels at the composition capabilities, while Terra excels at constructing a manifold.

---

> > ### Comment · Reviewer_STii · 2024-08-09
> > **Thank you for the response**
> >
> > Thanks the authors for their detailed response. The rebuttal addressed my concerns very well. I am happy to raise my scores. I think this is a technically strong paper with solid theoretical and empirical results, and it is is ready to be accepted.

---

### Official Review · Reviewer_LKTK · 2024-07-13

**Soundness:** 3
**Presentation:** 3
**Contribution:** 3
**Rating:** 6
**Confidence:** 3

**Summary:**

The paper presents a variant of LoRA, with additional time variable conditioned low-rank square matrix, for fine-tuning a diffusion model for unsupervised domain adaptation and domain generalization. Besides, the paper also study how to better apply the proposed Terra on UDA and DG tasks. Compared to prior arts, the proposed method achieve better performance.

**Strengths:**

- The paper presents an interesting idea on introducing a "time" condition on the LoRA matrix, which paves a path from source domain towards target domain.

- The paper also presents in depth analysis, providing interesting insight


- The paper is generally well-written and technically sound

**Weaknesses:**

- I feel the term "time" and "t" contradicts to the widely used time step t in diffusion model. Although I do see the authors used another symbol for diffusion models' timestep, it would be much better to use another term for the proposed "time" term to avoid confusion.

-  I appreciate the provided possible forms of Terra in Tab 6. Yet, I am wondering if the authors can provide more insights/principles in terms of how to choose the function form. Besides, there seems missing an ablation on the different functions for Terra, to justify why "Linear" is used for generative interpolation/UDA and "cosine-sine" is used for DG?

**Questions:**

Generally I feel the paper propose a simple yet interesting method for the domain generalization problem. Please see weakness for my questions.

**Limitations:**

No other limitation as far as I can see.

---

> ### Author Rebuttal · Authors · 2024-08-07
>
> #### [W.1] I feel the term "time" and "t" contradicts to the widely used time step t in diffusion model. Although I do see the authors used another symbol for diffusion models' timestep, it would be much better to use another term for the proposed "time" term to avoid confusion.
> > We appreciate the reviewer's concern. As you mentioned, we have used the symbol $\tau$ to differentiate our notation from the timestep in diffusion models. We intend to retain the term "time" because Terra draws inspiration from multiple research fields, including **Fluid Dynamics** and **Control Theory**, where "time" and "t" are also commonly used notations.
> >
> > - In the context of fluid dynamics, for each time variable $t$, the matrix updates of Terra is $\Delta W(t)$, enabling a time-dependent velocity field $\frac{d}{d t} \Delta W(t)$. Terra constructs a "LoRA flow" in the parameter space based on the Lagrangian and Eulerian descriptions [r1]. Therefore, after the cross-domain diffusion fine-tuning, Terra demonstrates the ability to generate domain flow.
> >
> > - From a control theory perspective, Terra can be viewed as solutions to time-varying systems. For instance, consider a linear system $\dot x(t) = A x(t)$, where the "state vector" $x(t)$ can be seen as the continuous image feature across different domains, and a closed-form solution is $x(t) = e^{A(t)} x_0$ [r2]. Here is the "exponential" form Terra with the time-varying matrix function.
> >
> > To clarify, we will provide a more detailed explanation of $\tau$ used in the diffusion model. Specifically, "since $t$ in this paper refers to the time variable in Terra, we use $\tau$ here to represent the timestep $t$ in the diffusion model".
> >
> > [r1] Villani C. Optimal transport: old and new. Berlin: springer, 2009, Pages 26.
> >
> > [r2] Williams R L, Lawrence D A. Linear state-space control systems. John Wiley & Sons, 2007, Pages 52-55.
>
>
> #### [W.2a] I appreciate the provided possible forms of Terra in Tab 6. Yet, I am wondering if the authors can provide more insights/principles in terms of how to choose the function form.
> > We appreciate the reviewer's interest in the possible forms of Terra presented in Table 6. To provide more insights, we elaborate on the guiding principles behind the choice of the three forms:
> >
> > - **Linear**: The $tW+I$ is the simplest form, related to a straight and steady flow, which is sufficient for two domains according to Theorem 1 and 2. Its constant velocity of weight changes ensures smooth morphing and is suitable for simple interpolating between two domains under the UDA setting.
> >
> > - **Cosine-Sine**: This form is adopted because of the bounded range and non-linearity of trigonometric functions, preventing image collapse during generation and enabling a complex parameter manifold to capture relationships between multiple domains. We recommend using this form in complex scenarios, such as interpolating multiple domains in DG.
> >
> > - **Exponential**: $e^{tW} = I + \sum_{k=1}^{\infty} \frac{t^k}{k!} W^k$, implemented using `torch.matrix_exp`, also defines a smooth curve in a high-dimensional manifold. This form is more expressive and suitable for handling multiple domains, as it enables feature transformations as mentioned in our response to [W.1]. Notably, it related to three types of transformations: scalings, rotations, and shears [r3].
> >
> > [r3] Ronald N. Goldman, VII.3 - Transformations As Exponentials, Graphics Gems II, 1991, Pages 332-337.
>
> #### [W.2b] Besides, there seems missing an ablation on the different functions for Terra, to justify why "Linear" is used for generative interpolation/UDA and "cosine-sine" is used for DG?
> > We appreciate the reviewer's comment and would like to provide clarification on this point.
> > - For tasks involving only two domains, such as generative interpolation and UDA, the expressive abilities of the "Linear" and "cosine-sine" functions are equivalent. Specifically, when $t=0$, both terms of the middle matrix reduce to the identity matrix $I$, and when $t=1$, the learnable matrices are constrained equally.
> > - However, for tasks involving more domains, such as DG, we have conducted an empirical ablation study to investigate the effectiveness of the "cosine-sine" form compared to the "linear" form. The results, presented in Table 7 of the manuscript and reproduced below, demonstrate the superiority of the "cosine-sine" form on the PACS dataset in DG scenarios.
> > \begin{array}{lcccccl}
> \hline
>  \textbf{Method} & \text{A} &   \text{C} & \text{P} & \text{S} &   \text{Average} \newline
> \hline
> \text{Linear} & 87.47 & 80.17 & 97.85 & 77.16 & 85.66   \newline
> \text{Cosine-sine (dim=1)}  & 88.29 & \textbf{82.36} & 97.53 & 73.31 & 85.37   \newline
> \text{Cosine-sine (dim=2)}  & \textbf{89.51} & 79.66 & \textbf{98.20} & \textbf{78.64} & \textbf{86.50}   \newline
> \hline
> \end{array}

---

> > ### Comment · Reviewer_LKTK · 2024-08-11
> > **post-rebuttal**
> >
> > Thanks for the clarification. Most of my concerns are addressed, and I would like to keep my final rating as weak accept.

---

### Official Review · Reviewer_76UZ · 2024-07-28

**Soundness:** 3
**Presentation:** 3
**Contribution:** 2
**Rating:** 7
**Confidence:** 4

**Summary:**

This article introduces Terra, a simple time-varying low-rank adapter based on LoRA for domain flow generation. Terra efficiently bridges the source and target domains using a parameter-efficient method. By generating data with smaller domain shifts, Terra effectively improves performance in incorporation, UDA, and DG tasks.

**Strengths:**

- The writing is clear and easy to follow.
- The method provides an intuitive and effective approach to enhancing LoRA for domain flow generation, with a theoretical analysis of its expressive power.
- Terra shows promising results in interpolation tasks.
- The idea of generating data with smaller domain shifts to the training set is innovative and enhances model performance in UDA and DG.

**Weaknesses:**

- The method essentially adopts the LoRA approach and constructs a low-rank parameter manifold through F(W,t)=tW+I. This can be seen as an interpolation version of LoRA with fewer parameters, which might limit the novelty of the model.
- There is a lack of more direct comparative experiments, as mentioned in the Questions section.
- As a method that fine-tunes SD, the paper would benefit from directly evaluating the quality of generated images using different fine-tuning methods (such as those mentioned in lines 58-61).
- Lack of discussion of limitations and failure cases.

**Questions:**

This article uses Terra-finetuned SD XL to generate more training data for UDA and DG, which is quite similar to other works that enhance classifier performance through data generation (e.g., [1,2]). Both approaches leverage the pre-trained SD's prior knowledge to generate images as a form of data augmentation. Hence, I recommend adding a comparison with these methods to demonstrate Terra's design advantages, given that both utilize the SD prior. If direct integration into the current evaluation framework is not feasible, a simple approach could be to generate corresponding data augmentations by changing prompts, showcasing the improvements brought by the SD prior itself. Alternatively, I encourage the authors to illustrate the respective contributions of Terra and the SD prior to the performance boost through other reasonable means.

Additionally, I noticed a minor issue in Table 1: DGP is based on a GAN model, while Terra uses SD XL. Therefore, a direct numerical comparison is unfair. I suggest the authors include the base model for clarity.

References:

[1] Synthetic data from diffusion models improves imagenet classification.

[2] Active Generation for Image Classification

**Limitations:**

The authors didn't discuss the limitations in the main paper.

---

> ### Author Rebuttal · Authors · 2024-08-07
>
> #### [W.1] The method essentially adopts the LoRA approach and constructs a low-rank parameter manifold through F(W,t)=tW+I. This can be seen as an interpolation version of LoRA with fewer parameters.
>
> > Terra constructs a "LoRA flow" in the parameter space, and is NOT an interpolation version of LoRA with fewer parameters. Here are three key differences:
> >
> > - **Formulation**: $tW+I$ is just one instance of Terra. Other possible forms are listed in Table 6 of the Appendix, i.e., $\exp(t{W})$ and $\cos(t{W})$.
> >
> > - **Training**: Unlike LoRA interpolation, which needs training separate LoRAs for different domains, our method only needs to train one adapter for multiple domains. In our training, the middle time-varying matrix $\mathcal{K}(t)$ is domain-specific, while the matrices $W_{up}$ and $W_{down}$ are shared across domains, enabling domain-general knowledge learning.
> >
> > - **Application**: The interpolation between two domain is just one application of our Terra. More importantly, based on Terra, we can design effective frameworks for UDA and DG:
> >   - For UDA, our framework can learn the domain-general subjects and domain-specific styles due to the Terra structure, enabling the generation of target-like samples from both source samples and random noise, thereby reducing the domain gap.
> >   - For DG, Terra constructs a domain manifold in the parameter space, facilitating random interpolation to generate diverse samples, which enhances the model's generalization ability.
>
> #### [W.2] There is a lack of more direct comparative experiments, as mentioned in the Questions.
>
> > Please refer to our following response to [Q.1a].
>
>
> #### [W.3] As a method that fine-tunes SD, the paper would benefit from directly evaluating the quality of generated images using different fine-tuning methods (such as those mentioned in lines 58-61).
> >- The methods mentioned involve customized image generation tasks, such as image editing and multi-concept generation, focusing primarily on single-domain image generation. In contrast, our approach enhances the LoRA structure to create a continuous parameter manifold, allowing for **image generation across a domain flow**.
> >
> >- A related work is Diffmorpher, which trains two LoRAs on image pairs and introduces techniques for **generating a continuously interpolative sequence of images**, referred to as image morphing. We compare Diffmorpher with our method in the manuscript.
> >
> >- To demonstrate that the paper's potential, we conduct an additional experiment by replacing the two LoRAs with our proposed Terra. The combined method "Terra + Diffmorpher" yields improved FID and PPL scores. The qualitative and quantitative results are presented in Fig. r1 and Tab. r1 of the Rebuttal-PDF.
>
> #### [W.4] Lack of discussion of limitations and failure cases.
> > - **Limitations**: We have thoroughly discussed the limitations in Appendix D. Additionally, while we have adapted to downstream domains through fine-tuning, our model may still be influenced by the prior of the foundation model to some extent.
> >
> > - **Failure cases**: We acknowledge that a small number of generated images may exhibit poor quality due to the conflict between SD prior knowledge and the knowledge required for downstream tasks. We showcase some failure cases in Fig. r3 of the Rebuttal-PDF. However, the number of those poor-quality images is small, and it does not affect the overall performance of the model.
>
> #### [Q.1a] I recommend adding a comparison with data augmentation with SD's prior knowledge to demonstrate Terra's design advantages. Alternatively, I encourage the authors to illustrate the respective contributions of Terra and the SD prior to the performance boost through other reasonable means.
> > We appreciate the reviewer's suggestion to explore the prior of foundation models. To address this concern, we design several methods to synthesize data based on the SDXL model and evaluate their effectiveness on UDA tasks:
> > - (1) **SDXL (random)**: We use the prompt `A [CLASS]` to generate samples for each class, where [CLASS] denotes the placeholder for the label.
> > - (2) **SDXL (styles)**: We first use the prompt `Generate 50 prompts describing diverse styles for image generation` to ask GPT-4, and then use the prompt `A [CLASS], an everyday object in office and home, in the style of [STYLE]` to generate samples, where [STYLE] denotes the placeholder for style prompts generated by GPT-4 (e.g. "Classic", "Modern").
> > - (3) **SDXL (target)**: Based on (2), we use the name of target domain (e.g. "Clipart") to replace the [STYLE] as the new placeholder for exploring the SD prior on the target domain.
> > - (4) **SDXL (target styles)**: We use the prompt `Generate 50 prompts describing [TARGET] style for image generation` to ask GPT-4 and obtain more detailed style prompts for synthesis.
> > - (5) **SDXL (selected)**: Inspired by [2], we use a confidence-based activate learning method to filter out poor-qulity and misclassified samples generated in (4) and select valid samples.
> >
> > The comparison results on Office-Home for UDA are shown in Tab. r3 of the Rebuttal-PDF. Terra outperforms the comparison methods, indicating that despite the boost in accuracy from target style design and active learning, the prior knowledge is insufficient to align with the downstream tasks. This issue can be furthur mitigated through finetuning with Terra, which demonstrates Terra's design advantages.
>
> #### [Q.1b]: I noticed a minor issue in Table 1: DGP is based on a GAN model, while Terra uses SD XL. Therefore, a direct numerical comparison is unfair. I suggest the authors include the base model for clarity.
> > Thank you for pointing out this issue. Here, we compare with GAN-based model DGP and the stable diffusion-based methods DDIM, LoRA Interpolation, and DiffMorpher. In our revision, we will include the base models for clarity. You can find the updated Tab. r1 and Fig. r1 of the Rebuttal-PDF.

---

> > ### Comment · Reviewer_76UZ · 2024-08-09
> >
> > Thanks for the authors' detailed response! My concern is well-addressed and I will raise the scores.

---

### Official Review · Reviewer_Y8y3 · 2024-07-29

**Soundness:** 3
**Presentation:** 2
**Contribution:** 3
**Rating:** 7
**Confidence:** 4

**Summary:**

This paper proposes Terra, a time-varying low-rank adapter based on the Low-Rank Adapter (LoRA) method, designed to enable continuous domain shifts from one domain to another. The core idea is to incorporate a time-dependent function between the LoRA low-rank matrices, using time t to control the interpolation of generated samples between the source and target domains. Qualitative results demonstrate that Terra facilitates continuous changes when transferring images across domains. Quantitative experiments indicate that Terra can serve as a foundation for generalization-based unsupervised domain adaptation and domain generalization tasks, thereby improving performance.

**Strengths:**

- The proposed method is well-motivated, straightforward and sounded.
- The authors conduct systematically experiments to verify the usage of Terra in multiple domains (by combining with most of the off-the-shelf methods).
- The code is provided.

**Weaknesses:**

- The presentation can be improved. I suggest elaborate and provide more details on the section 3.3 and to move Fig. 7 in the appendix to the main paper, since this part should be the most crucial part as how it constructs the evolving visual domains and serve as the basis to the following applications. Besides, the implementation details of morphing between style and subject should also be explained here.

- The experimental comparisons should be improved. For Sec 4.2, instead of only simply stating that "following the setting employed in DiffMorpher", the authors should also provide qualitative comparisons. Terra also does not show performance improvement against DiffMorpher or LoRA Interp. in terms of PPL. As a result, for Sec 4.3 and 4.4, it is expected that off-the-shelf UDA/DG + Terra can improve against those without Terra, but the comparisons should include off-the-shelf UDA/DG + other similar morphing works (e.g. DiffMorpher) for a fair comparison with prior arts.

**Questions:**

Please refer to the weaknesses.

**Limitations:**

Limitations are discussed in the paper. There are no potential negative societal impact.

---

> ### Author Rebuttal · Authors · 2024-08-07
>
> #### [W.1] The presentation can be improved. I suggest elaborate and provide more details on sec. 3.3 and to move Fig. 7 in appendix to the main paper. The implementation details of morphing between style and subject should be explained here.
>
> >We will follow your suggestion by (a) relocating Fig. 7 to Sec. 3.3 and (b) enhancing the organization of Sec. 3.3 to better illustrate how to construct evolving visual domains.
> > - **Stage 1: Fine-tune the parameters of Terra** (i.e., $\Delta_\theta=W_{up}\cup W_{mid}\cup W_{down}$) using the loss function defined in Eq. (5), where the first part with $t=0$ uses source samples $\mathcal{D}_S$ and the second part with $t=1$ uses target samples $\mathcal{D}_T$.
> > - **Stage 2: Generate an intermediate domain** by (a) uniformly sampling $t$ from [0,1] and (b) inputting the text prompt and a random noise into the fine-tuned diffusion model corresponding to domain $t$ (i.e., $\theta_0 + \Delta W(t)$ where $\theta_0$ is the pre-trained diffusion model) for the backward process.
> >
> >Regarding the implementation details of generative interpolation tasks, including those involving morphing between image pairs, styles and objects:
> >  - We will follow the reviewer's suggestion to **introduce a new section** on Generative Interpolation via Terra between Secs. 3.3 and 3.4. This section will list **generative interpolation as one of the three concurrent applications** of the proposed domain flow generation framework of Terra. It will present the details originally provided in Line 241-256 and 604-614.
> >  - For convenience, we also provide the details here.
> >    - Morphing in image pairs: In Stage 1, we instantiate the loss function in Eq. (5) by setting (a) $\mathcal{D}_S$ to include **one image of the pair**, (b) $\mathcal{D}_T$ to include **the other image**, and (c\) the text prompt to describe the images. In Stage 2, we generate intermediate images by uniformly transiting $t$ from 0 to 1 using the same text prompt as in Stage 1; each value of $t$ results in an interpolated image.
> >    - Morphing in styles/objects: This differs from morphing in image pairs only in $\mathcal{D}_S$ and $\mathcal{D}_T$ used in Stage 1, which are **a group of images in one style/object and a group of images in another style/object**, respectively.
>
> #### [W.2a] The authors should provide qualitative comparisons.
>
> >We provide qualitative comparisons of our method with all other baselines in Fig. r1 of the Rebuttal-PDF, which shows that Terra could generate intermediate images that are visually smooth and natural.
>
> #### [W.2b] Terra does not show performance improvement over DiffMorpher or LoRA Interp. in PPL. Comparisons should include off-the-shelf UDA/DG+other similar morphing works for a fair comparison.
> >We'd like to humbly clarify the potential misunderstandings.
> >
> >First, Terra is a **general framework that constructs a continuous parameter manifold** and thus generates a domain flow. To illustrate, in this paper, we show the three representative applications of Terra, including image morphing, UDA, and DG.
> >
> >Second, image morphing, UDA, and DG are three parallel applications of Terra, so that **the roles of Terra in them differ**.
> > - In image morphing, similar to DiffMorpher or LoRA Interpolation, Terra generates a continuously interpolative sequence of images.
> > - In UDA, as shown in Fig. 2(a), Terra bridges the gap between the source and target domains by generating more target-like samples from source samples and random noise. Thus,
> >   - in UDA, image morphing, whose output is an interpolative sequence of images, cannot be directly applied into off-the-shelf UDA methods.
> >   - for a fair comparison with prior arts in Terra's effectiveness in **generating more target-like samples**, we add a comparison against **off-the-shelf UDA + direct generation of target style samples** using prompts on Office-Home:
> >     - **SDXL (target)**: We use the prompt `A [CLASS] in the style of [TARGET]` to generate samples, where [CLASS] and [TARGET] denote the placeholders for the label name and target domain name, respectively.
> >     - **SDXL (target styles)**: We prompt GPT-4 with `Generate 50 prompts describing [TARGET] style for image generation` to obtain more detailed style descriptions for synthesis.
> >
> >     The results in Tab. r3 of the Rebuttal-PDF verify Terra's effectiveness in generating more target-like samples.
> >  - In DG, as shown in Fig. 2(b), Terra expands the source domains by (a) learning a $t$ predictor that maps each source domain to [-1,1] and (b) randomly sampling values of $t$ to generate more domains.
> >    - Image morphing can be adapted to off-the-shelf DG methods by including interpolative sequences of all pairs of images from different source domains, despite being computationally expensive.
> >    - For a fair comparison in Terra's effectiveness in **expanding source domains that generalize better**, we include the comparison against **off-the-shelf DG + morphing works** on Office-Home. That is, we train a LoRA for each domain and adopt **LoRA Interp./DiffMorpher** to interpolate. The results in Tab. r2 of the Rebuttal-PDF verify Terra's effectiveness, since Terra interpolates between domains instead of images and thus better models distributions in two domains.
> >
> >Third, even on the image morphing, Terra outperforms the DiffMorpher, which specifically designed for morphing by customized techniques such as attention interpolation, adaptive normalization, and a new sampling schedule.
> > - Terra supports morphing of styles and objects, which DiffMorpher seems incapable of.
> > - Terra enjoys better rationality and fidelity (measured by FID) than DiffMorpher.
> > - Terra is competitive in smoothness and consistency (**measured by PPL**) with DiffMorpher, despite being simple and general without any specific design. Moreover, **equipped with those customized techniques used in DiffMorpher, Terra is even better than DiffMorhper** (see Tab. r1 of the Rebuttal-PDF).

---

> > ### Comment · Reviewer_Y8y3 · 2024-08-12
> >
> > I appreciate the authors' efforts in providing additional clarifications and experimental results to address all of my concerns and I am happy with these answers. I will raise my final score to accept.

---

### Author Rebuttal · Authors · 2024-08-07

Dear Reviewers and ACs,

We sincerely thank all the reviewers and ACs for your diligent efforts and high-quality reviews. If you have any additional question or require further clarification, please feel free to let us know. Your insights are highly valued.

We are delighted to note that reviewers find that:

- our method is innovative (`Reviewer 76UZ`), well-motivated, straightforward, and sounded (`Reviewers Y8y3 and LKTK`), with clear and easy-to-follow writing (`Reviewers 76UZ, LKTK, and STii`).
- our paper is providing an intuitive and effective approach to construct a continuous parameter manifold for domain flow generation (`Reviewers 76UZ, LKTK, and STii`) and synthesize data to help alleviate domain shifts (`Reviewers 76UZ, LKTK, and STii`).
- our method includes a theoretical analysis of its expressive power (`Reviewers 76UZ, LKTK, and STii`), achieving promising results on various benchmarks (`Reviewers Y8y3, 76UZ, and STii`), and is supported by reproducible results with provided code (`Reviewers Y8y3 and STii`).

In responce to your valuable suggestions, we have conducted additional experiments and included the new results in the supplementary Rebuttal-PDF for your convenience:

- **Figure r1 & Table r1**: We have added the **qualitative** and **quantitative** comparison results of image morphing (suggested by `Reviewer y8Y3`) and introduced a combined method "Terra + Diffmorpher" (suggested by `Reviewers y8Y3 and 76UZ`).
- **Figure r2**: Supplementary samples for qualitative evaluation of the image morphing tasks (suggested by `Reviewer STii`).
- **Figure r3**: Some failure cases in generated samples (suggested by `Reviewer 76UZ`).
- **Figure r4**: We present the learned time variables to provide rationale behind our generation approach (suggested by `Reviewer STii`).
- **Table r2**: A comparison with **morphing works** (LoRA Interp. and DiffMorpher) that expand source domains under the DG setting (suggested by `Reviewer y8Y3`).
- **Table r3**: A comparison with **target-like samples augmentation methods** using the SDXL prior under the UDA setting (suggested by `Reviewers 76UZ and y8Y3`).
- **Table r4**: A **comparative analysis** with two state-of-the-art baseline methods (suggested by `Reviewer STii`).

Finally, due to character limits, we have condensed some reviews from `Reviewers y8Y3, 76UZ, and STii` in our responses.

Best regards,

The Authors

---

### Decision · Program_Chairs · 2024-09-25

**Decision:**

Accept (poster)

**Comment:**

This paper has received consistent feedback from four reviewers, all of whom are in favor of acceptance. It introduces Terra, a time-varying low-rank adapter based on the Low-Rank Adapter  method, which enhances LoRA for domain flow generation. The approach is simple yet effective, and the related experiments are comprehensive. Therefore,  the AC  has decided to accept this paper.